# Glycine cleavage system determines the fate of pluripotent stem cells via the regulation of senescence and epigenetic modifications

Shengya Tian[1,2,*], Junru Feng[2,*], Yang Cao[2], Shengqi Shen[2], Yongping Cai[3], Dongdong Yang[2], Ronghui Yan[2], Lihua Wang[2], Huafeng Zhang[2], Xiuying Zhong[1], Ping Gao[1,2,4]

Metabolic remodelling has emerged as critical for stem cell pluripotency; however, the underlying mechanisms have yet to be fully elucidated. Here, we found that the glycine cleavage system (GCS) is highly activated to promote stem cell pluripotency and during somatic cell reprogramming. Mechanistically, we revealed that the expression of Gldc, a rate-limiting GCS enzyme regulated by Sox2 and Lin28A, facilitates this activation. We further found that the activated GCS catabolizes glycine to fuel H3K4me3 modification, thus promoting the expression of pluripotency genes. Moreover, the activated GCS helps to cleave excess glycine and prevents methylglyoxal accumulation, which stimulates senescence in stem cells and during reprogramming. Collectively, our results demonstrate a novel mechanism whereby GCS activation controls stem cell pluripotency by promoting H3K4me3 modification and preventing cellular senescence.

## Introduction

Pluripotent stem cells (PSCs), including embryonic stem cells (ESCs) and induced pluripotent stem cells (iPSCs), have the ability to self-renew indefinitely and to differentiate into almost any type of somatic cell (Takahashi & Yamanaka, 2006; Ying et al, 2008; Shi et al, 2017). PSCs possess a unique metabolic system that is intimately linked to their pluripotent state (Folmes et al, 2012; Panopoulos et al, 2012; Zhang et al, 2012a; Shyh-Chang & Daley, 2015). Accumulating evidence has documented that similar to many types of cancer cells, PSCs preferentially obtain energy by high rates of glycolysis rather than by the more efficient process of aerobic respiration. Enhanced glycolysis promotes ESC self-renewal and improves the reprogramming efficiency of both mouse and human fibroblasts (Kondoh et al, 2007; Varum et al, 2011; Prigione et al, 2014; Cao et al, 2015). Recent studies have reported that, in contrast to the classic portrayal of the Warburg effect, pluripotent cells also use the glycolysis product Acetyl-CoA (Ac-CoA) to sustain histone acetylation and an open chromatin structure, which is critical for pluripotency and differentiation (Moussaieff et al, 2015). In addition to favouring glycolysis, PSCs also possess a distinct amino acid metabolism. For instance, mouse ESCs have the ability to catabolize threonine by activating threonine dehydrogenase (Tdh) to maintain an advantageous metabolic state; thus, mouse ESCs are very sensitive to threonine restriction (Wang et al, 2009; Shyh-Chang et al, 2013). However, because of the loss-of-function mutation of the Tdh gene during evolution, human ESCs have no ability to catabolize threonine; hence, whether human ESCs could benefit from metabolic pathways similar to threonine metabolism remains unclear. Intriguingly, a recent study performed by Shiraki et al noted that human ESCs were highly dependent on methionine metabolism, as methionine deprivation reduced histone and DNA methylation (Shiraki et al, 2014). More recently, an elegant study by Zhang et al (2016) showed that LIN28A regulated the serine synthesis pathway (SSP) in PSCs (Zhang et al, 2016). Despite these important findings regarding amino acid metabolism in PSCs, the underlying mechanisms and significance of amino acid metabolism in stem cells remain to be further explored.

The glycine cleavage system (GCS) is a multienzyme complex consisting of four individual components: glycine decarboxylase (Gldc), aminomethyltransferase (Amt), glycine cleavage system protein H (Gcsh), and dihydrolipoamide dehydrogenase (Dld). Gldc, Amt, and Gcsh are functionally specific to the GCS, whereas Dld encodes a housekeeping enzyme. As the first step of glycine cleavage in mitochondria, Gldc binds to glycine and transfers an aminomethyl moiety to Gcsh to form an intermediate in which the carboxyl carbon is converted to $CO_2$. Subsequently, Amt catalyses

[1]School of Medicine and Institutes for Life Sciences, South China University of Technology, Guangzhou, China    [2]Hefei National Laboratory for Physical Sciences at Microscale, The Chinese Academy of Sciences Key Laboratory of Innate Immunity and Chronic Disease, School of Basic Medical Sciences, Division of Life Science and Medicine, University of Science and Technology of China, Hefei, China    [3]Department of Pathology, School of Medicine, Anhui Medical University, Hefei, China    [4]Guangzhou Regenerative Medicine and Health Guangdong Laboratory, Guangzhou, China

Correspondence: zhongxy@scut.edu.cn; pgao2@ustc.edu.cn
*Shengya Tian and Junru Feng contributed equally to this work

the release of $NH_3$ from the Gcsh-bound intermediate and transfers the methylene to tetrahydrofolate (THF), forming 5,10-methylene THF (Kikuchi, 1973; Narisawa et al, 2012; Go et al, 2014). The GCS is activated in only a few adult human tissues, mostly in the liver, brain, lung, and kidney, but its function in these tissues remains elusive (Kure et al, 2001). Inborn defects in GCS activity caused by mutations in Gldc or Amt lead to severe non-ketotic hyperglycinemia (NKH), which is life-threatening and leads to severe neurological disorders (Kikuchi et al, 2008; Pai et al, 2015; Leung et al, 2017). Recently, the GCS was found to be associated with many types of cancers; for example, GCS dys-regulation promotes non–small cell lung cancer as well as glioma (Zhang et al, 2012b; Kim et al, 2015). However, the GCS was also reported to suppress the progression of hepatocellular carcinoma by inhibiting cell invasion and intrahepatic metastasis (Zhuang et al, 2018). Collectively, these results highlight the cell context-dependent role of the GCS in cell fate determination. It is interesting to note that although the association of abnormal GCS activity with cancers and diseases has been appreciated, there is still little documentation showing the means by which the GCS is regulated. More intriguingly, although cancer cells share a variety of properties with stem cells, it is much less understood how the GCS functions in PSCs than in other cells. Notably, during the submission of this manuscript, a very recent article described the role of the GCS in the maintenance and induction of pluripotency via metabolic regulation (Kang et al, 2019). However, additional mechanistic details, as well as the significance of this finding, remain to be further explored.

Here, we found that the GCS is highly activated in PSCs as a result of the elevated expression of the rate-limiting enzyme Gldc. We demonstrate that Gldc is activated by Sox2 and Lin28A via transcriptional and posttranscriptional mechanisms, respectively. We further demonstrate that the activated GCS is critical for the PSC pluripotency via the maintenance of the epigenetic state in PSCs as well as the protection of PSCs from senescence. Taken together, our findings illustrate a novel mechanism by which the GCS contributes to the unique metabolic phenotype of stem cells to control their pluripotency and cell fate.

# Results

## Gldc-mediated GCS is activated in PSCs and during somatic cell reprogramming

To investigate whether ESCs exhibit a metabolic state different from that of somatic cells, we performed RNA-seq analysis of MEFs, iPSCs, and mouse embryonic stem cells (mESCs). Hierarchical clustering analysis of metabolic genes revealed that iPSCs and mESCs had a similar expression pattern, which was distinct from that of MEFs (Fig 1A). Gene ontology term enrichment analysis demonstrated that genes involved in the metabolism of glycine, serine, and threonine as well as carbon were significantly up-regulated in stem cells (Fig 1B). Consistent with previous reports (Wang et al, 2009; Shyh-Chang et al, 2013), our data also showed that Tdh, the enzyme that catalyses threonine oxidation, was up-regulated in iPSCs and mESCs (Figs 1C and S1A). Interestingly, Gldc, the key enzyme in the GCS, exhibited markedly elevated expression in both iPSCs and mESCs compared with that in MEFs (Figs 1C and S1A). qRT-PCR

analysis of the genes involved in glycine, serine, and threonine metabolism revealed that many of the metabolic enzymes responsible for glycine metabolism, such as Gldc, Amt, Tdh, and Gcat, were much more abundant in iPSCs and mESCs than in MEFs (Fig 1D and E). Notably, the expression of Gldc in iPSCs and mESCs was more than 200 times higher than that in MEFs (Fig 1E). Consistent with these findings, the Western blot analysis results further revealed that the protein levels of glycine dehydrogenase (GLDC), amino-methyltransferase (AMT), L-threonine dehydrogenase (TDH), and glycine C-acetyltransferase (GCAT) were significantly higher in iPSCs and mESCs than in MEFs, with GLDC exhibiting the greatest difference (Fig 1F). More importantly, during somatic cell reprogramming of MEF cells by the four Yamanaka factors (Oct4, Sox2, Klf4, and c-Myc), the expression of Gldc gradually increased, even before the induction of endogenous pluripotency genes, suggesting that Gldc may play an important role in iPSC generation (Figs 1G and S1B). Similar results were also observed during reprogramming induced by three of the Yamanaka factors (Oct4, Sox2, and Klf4) or the Thomson factors (Oct4, Sox2, Nanog, and Lin28A) (Fig S1C and D). On the other hand, the protein level of GLDC was gradually decreased in V6.5 cells during retinoic acid–induced differentiation (Fig S1E). Collectively, these data strongly suggest that Gldc is associated with pluripotency acquisition and maintenance.

Measurement of GCS enzymatic activity by a glycine decarboxylation assay showed that both iPSCs and mESCs exhibited significantly higher GCS activity than MEFs (Fig 1H). Moreover, the activity of the GCS gradually increased during reprogramming (Fig 1I). Further tracing experiments using [13]C-labelled glycine revealed that iPSCs and mESCs consumed much more extracellular [13]C-labelled glycine than MEFs, suggesting greater glycine utilization by PSCs than by somatic cells (Fig 1J). More importantly, knockdown of Gldc in V6.5 cells significantly decreased the activity of the GCS (Fig 1K), suggesting that the induced Gldc expression is responsible for the activation of the GCS in mESCs and iPSCs. Taken together, these data demonstrate that the Gldc-mediated GCS is activated in PSCs and during somatic cell reprogramming.

## Sox2 and Lin28A regulate Gldc expression via distinct mechanisms

To explore the underlying mechanisms of Gldc regulation, we individually overexpressed the Yamanaka factors and Thomson factors in MEF cells to assess the expression of Gldc. Our results revealed that overexpression of Sox2 markedly elevated the Gldc mRNA and protein levels, whereas Lin28A enhanced the protein but not the mRNA expression (Fig 2A). The protein level of Gldc was also mildly increased in Klf4 overexpressing MEFs (Fig 2A), which is consistent with Kang's study (Kang et al, 2019). Furthermore, knockdown of Sox2 with shRNAs markedly decreased both the mRNA and protein levels of Gldc in V6.5 cells (Fig 2B), indicating that Sox2 is critical for the transcriptional activation of Gldc. Analysis of the publicly available ChIP-seq data for Sox2 in mESCs (Whyte, 2013) (GSE44288) revealed that Sox2 was enriched in the promoter region of Gldc (Fig S2A). In addition, our ChIP experiment showed that Sox2 binds to the promoter region of the Gldc gene, indicating that Gldc is a direct transcriptional target of Sox2 (Fig 2C). We then performed a dual luciferase assay using reporter plasmids containing

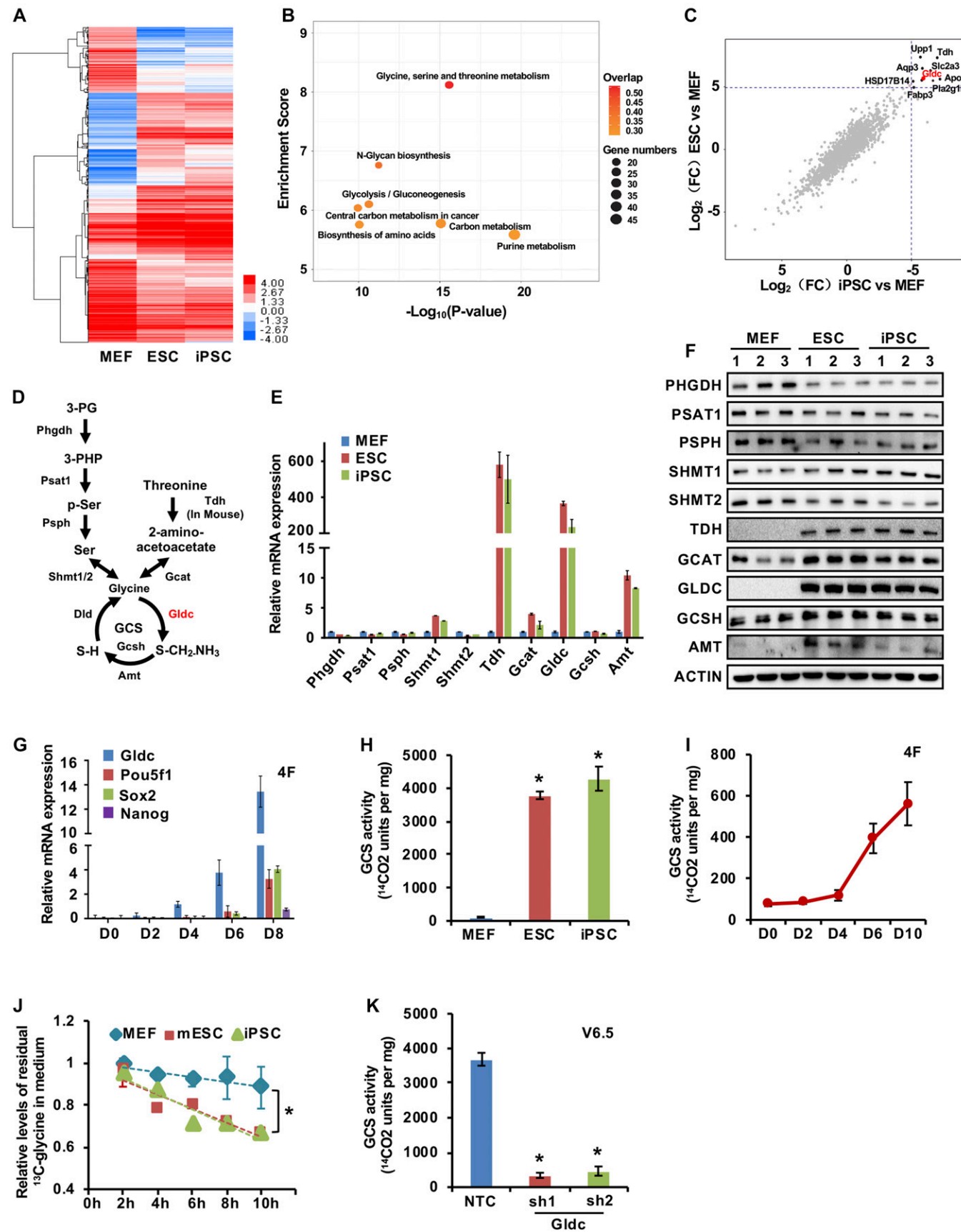

potential Sox2-binding sites as well as the corresponding mutated fragment (Fig 2D). Sox2 significantly enhanced the luciferase activity of the reporter containing the wild-type Gldc promoter but not the reporter containing the mutated fragment (Fig 2E). Thus, these data demonstrate that Sox2 directly binds to the promoter of Gldc to activate its transcription.

Western blot and qRT-PCR analyses revealed that knockdown of Lin28A by shRNAs significantly decreased the protein level of GLDC but not its mRNA level in V6.5 cells (Figs 2F and S2B), suggesting that Lin28A is involved in the posttranscriptional regulation of Gldc. To explore whether Lin28A regulates Gldc through let-7, we introduced Dgcr8$^{-/-}$ mESCs, which lack the expression of mature miRNAs (Wang et al, 2007). Western blot analysis revealed that in Dgcr8-depleted mESCs, forced expression of Lin28A enhanced GLDC protein expression (Fig S2C), suggesting that Lin28A regulates GLDC through let-7–independent mechanisms. To explore whether Lin28A functions as an RNA-binding protein, we performed the RNA immunoprecipitation (RIP) assay in V6.5 cells with a Lin28A-specific antibody, and the results demonstrated that Lin28A directly binds to Gldc mRNA (Fig 2G). As previously reported, the cold-shock domain and CCHC (retroviral-type CCHC zinc knuckles) domains are important for Lin28A binding to RNA (Piskounova et al, 2008; Balzer & Moss, 2014); thus, we then constructed a plasmid containing mutated Lin28A by replacing three conserved surface aromatic residues in the cold-shock domain and two conserved histidines in the CCHC domain with alanines to generate a mutated LIN28A protein that is deficient in RNA binding (Fig S2D). Our data showed that this mutated Lin28A showed no promotive effect on GLDC protein expression by Western blotting and no binding to Gldc mRNA by RIP assay, indicating that Lin28A promotes the expression of Gldc as an RNA-binding protein (Fig 2H and I). To further investigate the precise Lin28A-binding regions in Gldc mRNA, publicly available Lin28A crosslinking-immunoprecipitation-seq data from mESCs were analysed (Cho et al, 2012) (GSE37111), and the results revealed two potential Lin28A-binding regions in Gldc mRNA (Fig S2E). Next, we cloned those two fragments (region 1 and region 2) containing potential Lin28A-binding sites as well as a negative control region (region 3) of Gldc mRNA into a reporter vector (Fig 2J). Dual luciferase assay revealed that Lin28A significantly enhanced the luciferase activity of the reporters containing wild-type Gldc, Gldc region 1, or Gldc region 2, but not that of the reporter containing region 3 (Fig 2K). Interestingly, deletion of region 1 or region 2 attenuated the Lin28A-enhanced reporter activity driven by wild-type Gldc, and the

double mutation completely abolished the enhanced luciferase activity (Fig 2L), suggesting that Lin28A binds to Gldc mRNA via region 1 and region 2 to enhance Gldc protein expression.

Consistent with these results, knockdown of either Sox2 or Lin28A decreased the expression of Gldc and the enzymatic activity of the GCS, both of which were further reduced when Sox2 and Lin28A were knocked down simultaneously (Fig 2M). Overall, these data suggest that Sox2 and Lin28A promote the expression of Gldc via distinct mechanisms.

## GCS promotes one-carbon unit production and attenuates methylglyoxal (MG) accumulation

To determine whether the activation of Gldc altered the metabolism of mESCs, we generated V6.5 cells stably expressing shGldc to identify the cellular metabolites involved in glycine metabolism (Figs 3A and S3A). Our gas chromatography/liquid chromatography-mass spectrometer (GC/LC-MS) data showed that a large number of cellular metabolites were altered by shGldc. In particular, the levels of metabolites of one-carbon metabolism, including formate, 5mTHF, S-adenosyl methionine (SAM), and AMP, were significantly decreased, whereas cellular glycine markedly accumulated (Fig 3B). Excess glycine can be converted to serine by Shmts, providing materials for the biosynthesis of glutathione (GSH), or can be converted into 2-amino-3-ketobutyrate by glycine acetyltransferase (Gcat), which can be further catabolized into MG (Fig 3A). Our data showed that suppression of Gldc had no significant effect on the cellular levels of serine, threonine, or GSH but led to cellular accumulation of MG (Fig 3B and C). Next, we performed the metabolic flux tracing experiment using $^{13}$C-labelled glycine by LC/GC-MS. Our results revealed that knockdown of Gldc decreased the cellular levels of $^{13}$C-labelled formate and SAM in V6.5 cells (Fig 3D and E), suggesting that Gldc is essential for the generation of one-carbon units from glycine. Furthermore, when Gldc expression was knocked down during 4-factor–induced reprogramming, the cellular levels of formate decreased, whereas MG accumulated (Fig 3F and G). Taken together, these data demonstrate that Gldc is critical for promoting one-carbon units production and preventing MG accumulation in pluripotent cells.

Somatic cell reprogramming experiments using MEFs induced by the four Yamanaka factors revealed that suppression of Gldc in MEF cells markedly inhibited the formation of alkaline phosphatase AP$^+$/Ssea1$^+$ colonies (Fig 3H). However, overexpression of Gldc

**Figure 1. The Gldc-mediated GCS is activated in PSCs and during somatic cell reprogramming.**
**(A)** The heat map from the RNA-seq analysis showed alterations in the expression of metabolic genes in mESCs and iPSCs relative to the expression of these genes in MEFs. The colours indicate the ln-transformed FPKM values. **(B)** Gene ontology term enrichment was assessed using the algorithm on the DAVID website. **(C)** RNA-seq analysis results comparing mESCs with MEFs and comparing iPSCs with MEFs. Each point represents the log$_2$-transformed fold change for a given transcript. Genes significantly up-regulated in mESCs and iPSCs are annotated. **(D)** Schematic diagram of the glycine, serine, and threonine metabolism pathway. **(E, F)** qRT-PCR (E) and Western blot (F) analyses of the expression of the enzymes in glycine, serine, and threonine metabolic pathway in MEFs, mESCs, and iPSCs. ACTIN served as the loading control. The data were presented as the mean ± SD of three independent experiments. *P < 0.05 compared with MEFs; t test. **(G)** qRT-PCR analysis of the expression of Gldc, endogenous Pou5f1, Sox2, and Nanog during reprogramming at the indicated times. The mRNA levels were normalized to the expression levels on day 0. *P < 0.05 compared with the expression levels on day 0; t test. **(H)** The enzymatic activity of the GCS was measured in MEFs, mESCs, and iPSCs. The data were presented as the mean ± SD of three independent experiments. *P < 0.05 compared with MEFs; t test. **(I)** The enzymatic activity of the GCS was measured during reprogramming for the indicated times. The data were presented as the mean ± SD. *P < 0.05 compared with the activity on day 0; t test. **(J)** MEFs, mESCs, and iPSCs were first cultured in glycine starvation medium for 12 h, and the medium was then refreshed with medium containing $^{13}$C-labelled glycine. The amount of $^{13}$C-labelled glycine in the culture medium from MEFs, iPSCs, and mESCs was measured by GC/MS at the indicated time. The data were presented as the mean ± SD of three independent experiments. *P < 0.05 compared with MEFs; t test. **(K)** The enzymatic activity of the GCS was measured in V6.5 cells stably expressing shGldc or nontargeting control (NTC) shRNAs. The data were presented as the mean ± SD. *P < 0.05 compared with the NTC; t test.
Source data are available for this figure.

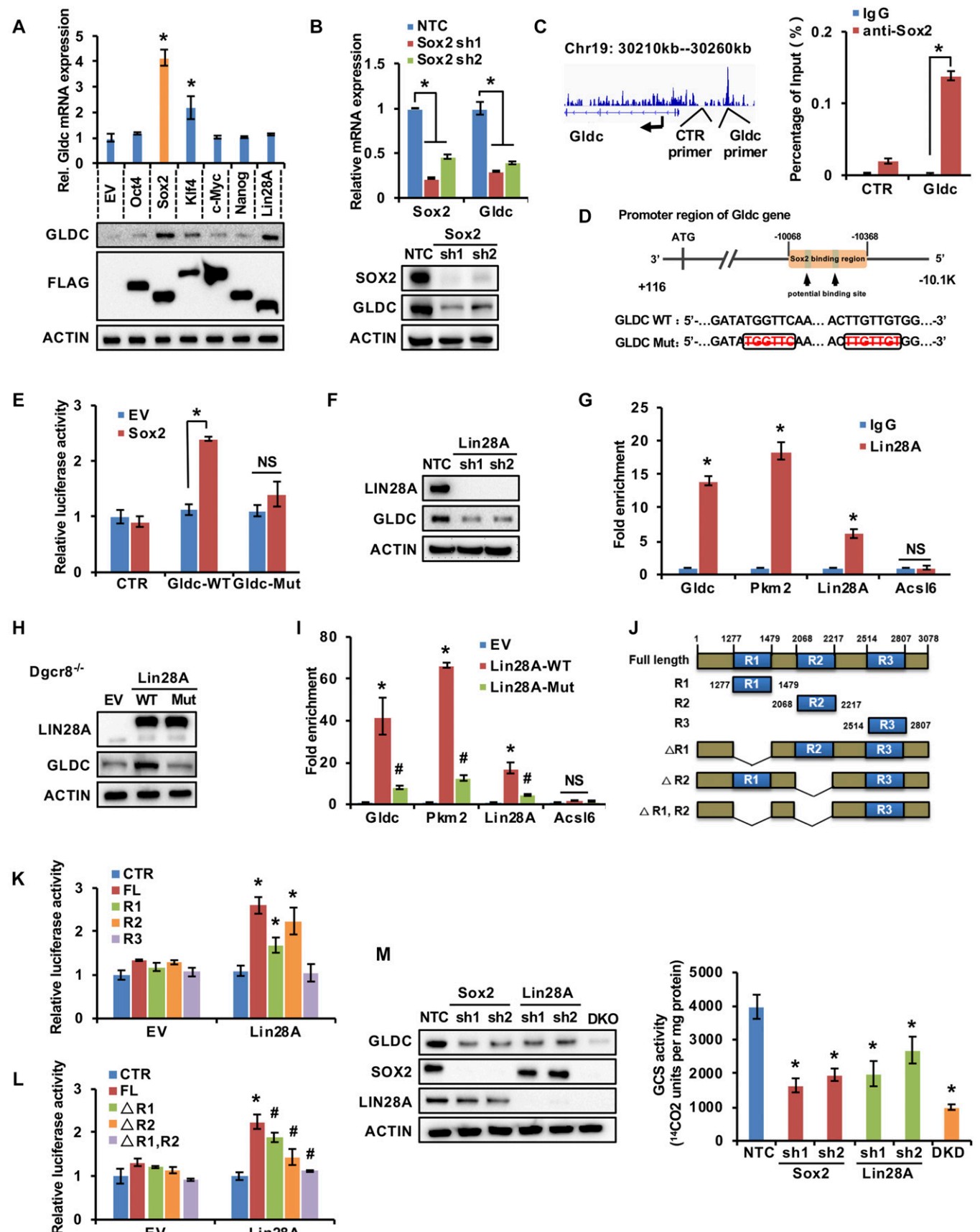

markedly increased the number of iPSC colonies by several fold in cells undergoing both 4-factor- and 3-factor–induced reprogramming (Figs 3I and S3B), suggesting that Gldc is critical for somatic cell reprogramming. The results of the staining experiments for stem cell markers (Oct4, Sox2, Ssea1, and Nanog) as well as the in vivo teratoma formation experiments confirmed the pluripotency of these iPSCs (Fig 3J and K). More importantly, supplementation with formate, a central one-carbon metabolite in the THF cycle, partially restored the iPSC formation suppressed by shGldc (Fig 3L). However, intriguingly, MG supplementation abolished the enhancement of reprogramming efficiency induced by Gldc overexpression (Fig 3M). Collectively, these data demonstrate that both one-carbon metabolites and MG are involved in Gldc-mediated somatic cell reprogramming.

### GCS promotes the pluripotency of stem cells by regulating H3K4me3

To further investigate the effect of Gldc in mESCs, we analysed our RNA-seq data from V6.5 cells with Gldc knockdown. Gene set enrichment analysis (GSEA) showed that the expression levels of genes associated with stem cell pluripotency were decreased significantly in Gldc knockdown cells, whereas the house keeping genes were not changed (Figs 4A and S4A). qRT-PCR analysis confirmed that Gldc knockdown markedly suppressed the expression of pluripotency genes in V6.5 cells as well as in 4F-induced iPSCs (Figs 4B and S4B). Western blot analysis with available antibodies against OCT4, SOX2, KLF4, and NANOG also confirmed the regulation of those genes by Gldc in V6.5 cells (Fig 4C). More interestingly, consistent with the data in Fig 3 indicating that Gldc regulates one-carbon units, GSEA analysis of our RNA-seq data showed that suppression of Gldc significantly inhibited one-carbon metabolic pathways (Fig 4D). Alterations in cellular one-carbon units have been shown to regulate histone methylation and subsequent gene expression (Shyh-Chang et al, 2013). Our Western blot analysis also showed that histone 3 lysine 4 trimethylation (H3K4me3) was significantly reduced when Gldc was knocked down in both V6.5 cells and 4F-induced iPSCs (Figs 4E and S4C). ChIP assay

using anti-H3K4me3 antibody revealed that the H3K4me3 levels on the transcription starting site of pluripotency genes were significantly reduced by shGldc in V6.5 cells as well as in 4F-induced iPSCs (Figs 4F and S4D), suggesting that Gldc plays an important role in maintaining the stem cell fate of mESCs via epigenetic reprogramming.

To evaluate the contribution of one-carbon units to Gldc deficiency-induced loss of pluripotency, we added 1 mM formate to the culture medium, and the Western blot results showed that supplementation with exogenous formate restored H3K4me3 suppressed by shGldc in V6.5 cells as well as in 4F-induced iPSCs (Figs 4G and S4E). Moreover, ChIP assays demonstrated that supplementation with exogenous formate partially restored the shGldc-suppressed H3K4me3 levels on the transcription start site of pluripotency genes in V6.5 cells (Fig 4H). Consistent with these findings, the expression of pluripotency genes suppressed by shGldc was partially rescued by formate supplementation (Fig 4I and J). Interestingly, in PSCs and during somatic reprogramming, the up-regulated Sox2 transcriptionally induced the expression of Gldc (Fig 2A and B), which promotes the one-carbon metabolism to maintain H3K4me3 on pluripotency-related genes, including Sox2 (Fig 4F). These data suggested a positive feedback loop between Gldc and Sox2, which is critical for acquisition and maintenance of pluripotency.

To test whether Gldc-mediated histone methylation plays an important role in somatic cell reprogramming, we treated MEF cells with two H3K4 methyltransferase inhibitors during reprogramming. Both inhibitors markedly attenuated the promotive effects of Gldc on reprogramming (Fig 4K). Furthermore, supplementation with exogenous SAM iodide partially rescued the shGldc-suppressed reprogramming efficiency (Fig 4L), suggesting that a defect in one-carbon metabolism is one of the major mechanisms underlying the shGldc-mediated suppression of stem cell pluripotency. Taken together, these data suggest that one-carbon metabolism and subsequent histone methylation are involved in Gldc-mediated pluripotency regulation.

### The GCS protects PSCs from MG-induced senescence

As shown in Fig 3, we observed that knockdown of Gldc resulted in the accumulation of MG. MG is a toxic, highly reactive aldehyde that

**Figure 2. Sox2 and Lin28A cooperatively control Gldc expression.**
**(A)** qRT-PCR and Western blot analysis of the expression of Gldc in MEF cells overexpressing Oct4, Sox2, Klf4, c-Myc, Nanog, and Lin28A individually. ACTIN served as the loading control. The data were presented as the mean ± SD of three independent experiments. *$P$ < 0.05 compared with the EV control; $t$ test. **(B)** qRT-PCR and Western blot analysis of the expression of Gldc in V6.5 cells stably expressing shSox2 or the NTC. The data were presented as the mean ± SD of three independent experiments. *$P$ < 0.05 compared with the NTC; $t$ test. ACTIN served as the loading control. **(C)** ChIP-qPCR analysis of potential binding site occupancy by Sox2 in the Gldc promoter in V6.5 cells using IgG or anti-Sox2 antibody. The data were presented as the mean ± SD. *$P$ < 0.05 compared with the IgG control; $t$ test. **(D)** Schematic diagram showing the Sox2-binding fragment in the Gldc promoter. **(E)** HEK293 cells were cotransfected with pSIN-3×flag-Sox2 and pGL2-P-Gldc wild-type, pGL2-P-Gldc mutant, or pGL2-P-EV luciferase vectors. A dual luciferase assay was performed 48 h after transfection. The data were presented as the mean ± SD. *$P$ < 0.05 compared between the indicated groups; $t$ test. **(F)** Western blot analysed the expression of GLDC in V6.5 cells stably expressing shLin28A or NTC. ACTIN served as loading control. **(G)** RIP assay followed by qPCR analysed the interaction between Gldc mRNA and Lin28A. Pkm2 served as positive control, Acsl6 served as negative control. Data were presented as the mean ± SD. *$P$ < 0.05 compared with IgG; $t$ test. **(H)** Western blot analysis of GLDC expression in Dgcr8$^{-/-}$ V6.5 cells stably expressing Lin28A wild-type. ACTIN served as the loading control. **(I)** RIP assay was performed in V6.5 cells transfected with pSIN-3×flag-Lin28A wild-type, pSIN-3×flag-Lin28A mutant, or pSIN-3×flag-EV using anti-Flag antibody, followed by qPCR to investigate the interaction between Gldc mRNA and Lin28A. Pkm2 served as the positive control and Acsl6 served as the negative control. The data were presented as the mean ± SD. *$P$ < 0.05 compared with empty vector; #$P$ < 0.05 compared with Lin28A wild-type; $t$ test. **(J)** Mutants of Gldc used in the dual luciferase assay. **(K)** HEK293 cells were cotransfected with pSIN-3×flag-Lin28A and either pSI-check2-Gldc region 1/2/3, full-length Gldc, or pSI-check2-EV luciferase vectors. A dual luciferase assay was performed 48 h after transfection. The data were presented as the mean ± SD. *$P$ < 0.05 compared between CTR; $t$ test. **(L)** HEK293 cells were cotransfected with pSIN-3×flag-Lin28A and either pSI-check2-Gldc full-length, pSI-check2-Gldc full-length with region 1/2 deleted, or pSI-check2-EV luciferase vectors. A dual luciferase assay was performed 48 h after transfection. The data were presented as the mean ± SD. *$P$ < 0.05 compared between the indicated groups, #$P$ < 0.05 compared with pSI-check2-Gldc full-length; $t$ test. **(M)** V6.5 cells were infected with viruses expressing shSox2 and shLin28A, followed by analysis of GLDC expression by Western blotting (left) and analysis of GCS enzymatic activity (right). ACTIN served as the loading control. The data were presented as the mean ± SD. *$P$ < 0.05 compared with the NTC; $t$ test. Source data are available for this figure.

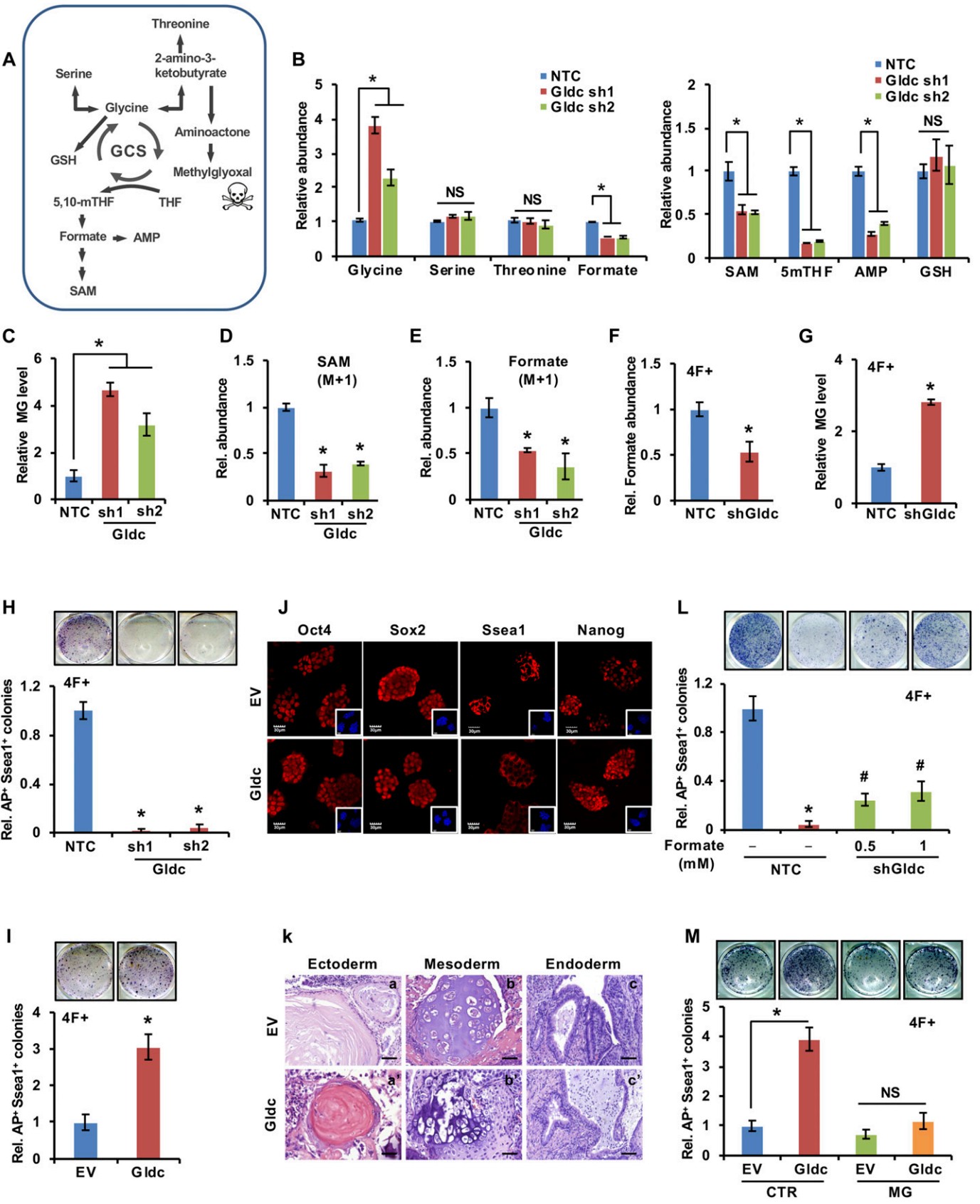

can react with lipids, nucleic acids, or with lysine and arginine residues of proteins to form advanced glycation end (AGE) products (Chaudhuri et al, 2018). MG has been reported to be associated with the pathology of age-related neurodegenerative disease and other disorders and accumulates during ageing (Kuhla et al, 2005; Matafome et al, 2012; Allaman et al, 2015). Intriguingly, we observed that MG treatment dose-dependently induced the expression of senescence markers in V6.5 cells (Fig 5A), suggesting that MG may lead to ESC senescence. During MEF cell reprogramming induced by the four Yamanaka factors, supplementation with MG led to increased levels of argpyrimidine, an MG-derived AGE, and senescence markers, as well as to the accumulation of $\beta$-gal–positive cells (Fig 5B and C). AP and Ssea1 staining revealed that the addition of MG decreased the number of iPSC colonies in a dose-dependent manner (Fig 5D), suggesting that MG reduces the reprogramming efficiency. Consistent with the data in Fig 3 indicating that suppression of Gldc led to the accumulation of MG, the Western blotting results showed that suppression of Gldc in V6.5 cells increased the cellular levels of argpyrimidine, an MG-derived AGE (Figs 5E and S5A). More importantly, knockdown of Gldc in V6.5 cells induced the expression of senescence markers (Fig 5E). During reprogramming, Gldc suppression also resulted in increased levels of argpyrimidine and senescence markers, as well as increased $\beta$-gal–positive cells (Figs 5F and G, and S5B).

Previous studies showed that MG could be detoxified to D-lactate by the glutathione-dependent glyoxalase system (Ahmed et al, 2003) and that carnosine could protect proteins against MG-mediated modifications (Hipkiss, 1998). Thus, we next investigated the involvement of MG in Gldc deficiency-induced senescence by using carnosine or by manipulating Glo1, an MG detoxifier in the glyoxalase system. Supplementation with carnosine or forced expression of Glo1 in V6.5 cells attenuated shGldc-induced mRNA expression of the senescence markers (Fig 5H). The Western blot results revealed that overexpression of Glo1 attenuated the shGldc-induced increase in the argpyrimidine level as well as the protein levels of the senescence markers in V6.5 cells (Fig 5I). Similar results were observed during reprogramming of MEF cells (Fig 5J). $\beta$-Gal staining further confirmed that Glo1 attenuated the senescence induced by shGldc during reprogramming (Fig 5K), suggesting that MG accumulation is involved in Gldc deficiency-induced senescence. AP and

Ssea1 staining revealed that forced expression of Glo1 or supplementation with carnosine partially recovered shGldc-suppressed reprogramming efficiency (Figs 5L and S5C). These data suggest that MG clearance, in addition to one-carbon unit production, directly modulates Gldc-mediated reprogramming efficiency.

Taking these two mechanisms into consideration, we performed the reprogramming experiment with combined treatment with SAM supplementation and forced Glo1 overexpression. Consistent with the above results, we observed that either ectopic Glo1 expression or SAM supplementation partially recovered the reprogramming efficiency reduced by shGldc and that this recovery was further potentiated by combined treatment with SAM and Glo1 overexpression (Fig 5M). Collectively, our data show that Gldc promotes the pluripotency of stem cells through at least two mechanisms: up-regulating the expression of pluripotency genes through epigenetic reprogramming and preventing senescence caused by toxic MG accumulation.

### GCS activation is conserved in human ESCs/iPSCs

Next, we tested whether GCS activation and the underlying mechanism were conserved in human cells. Consistent with our results from the mouse system, the qRT-PCR and Western blot analysis results showed that GLDC expression was dramatically increased in human ESCs (Fig 6A and B). Notably, the expression of enzymes in the SSP was also greatly increased (Fig 6A and B), a pattern distinct from that in mESCs (Fig 1E and F). It seems that, as TDH is a pseudogene in humans, human ESCs are deficient in catabolizing threonine toward glycine; thus, these cells switch to a SSP to provide enough glycine from serine for cells to use. Furthermore, enzyme activity analysis showed that human H9 ESCs have much higher GCS activity than the human fibroblast IMR90 cells (Fig 6C). During the reprogramming of human IMR90 cells by the four Yamanaka factors (OCT4, SOX2, KLF4, and c-MYC), GLDC expression was also gradually up-regulated (Fig 6D). Consistent with the data in mESCs, knockdown of SOX2 or LIN28A markedly reduced the expression of GLDC in H9 cells (Fig 6E and F), suggesting that SOX2- and LIN28A-mediated GCS activation is conserved in human ESCs. Knockdown of GLDC in human H9 cells significantly decreased the level of H3K4me3 and increased the cellular level of

**Figure 3. GCS promotes the production of one-carbon units and prevents the accumulation of MG.**
**(A)** Schematic diagram of metabolic flux related to the GCS. **(B)** Metabolites related to the GCS were measured by LC/GC-MS in V6.5 cells stably expressing shGldc or the NTC. The data were presented as the mean ± SD. *$P$ < 0.05 compared with the NTC; $t$ test. **(C)** MG measurement with biological kit in V6.5 cells stably expressing shGldc or the NTC. The data were presented as the mean ± SD. *$P$ < 0.05 compared with the NTC; $t$ test. **(D)** LC-MS analysis of the production of $^{13}$C-labelled SAM from $^{13}$C-labelled glycine in V6.5 cells stably expressing shGldc or the NTC. The data were presented as the mean ± SD. *$P$ < 0.05 compared with the NTC; $t$ test. **(E)** GC-MS analysis of the production of $^{13}$C-labelled formate from $^{13}$C-labelled glycine in V6.5 cells stably expressing shGldc or the NTC. The data were presented as the mean ± SD. *$P$ < 0.05 compared with the NTC; $t$ test. **(F)** GC-MS analysis of the formate content in reprogramming cells expressing shGldc or the NTC. The data were presented as the mean ± SD. *$P$ < 0.05 compared with the NTC; $t$ test. **(G)** Cellular MG levels were measured in reprogramming cells expressing shGldc or the NTC. The data were presented as the mean ± SD. *$P$ < 0.05 compared with the NTC; $t$ test. **(H, I)** MEF cells with Gldc knockdown (H) or overexpression (I) were infected with virus expressing the four factors to induce iPSC formation. AP staining (upper panel) showed the iPSC colonies formed. The AP- and Ssea1-positive iPSC colonies were counted (lower panel). The data were presented as the mean ± SD. *$P$ < 0.05 compared with the NTC/EV; $t$ test. **(J)** Immunofluorescence analysis was performed in iPSCs generated from the experiment shown in (I) using anti-Oct-4/-Sox2/-Ssea1/-Nanog antibodies. Nuclei were visualized by DAPI staining. **(K)** Haematoxylin and eosin staining of teratoma sections derived from the iPSCs generated from the experiment shown in (I) showing the differentiation of iPSCs into cell types of the three germ layers: keratinous epithelium (a, a'), cartilage (b, b'), and glandular tissue (c, c'). Scale bars, 50 $\mu$m. **(L)** Formate was added to the medium of MEF cells with Gldc knockdown starting from 2 d after infection with virus expressing the four factors. AP staining (upper panel) showed the iPSC colonies formed. The AP- and Ssea1-positive iPSC colonies were counted (lower panel). The data were presented as the mean ± SD. *$P$ < 0.05 compared with the NTC. #$P$ < 0.05 compared with the Gldc knockdown group; $t$ test. **(M)** MG was added to the medium of Gldc-overexpressing MEF cells starting from 2 d after infection with virus expressing the four factors. AP staining (upper panel) showed the iPSC colonies formed. The AP- and Ssea1-positive iPSC colonies were counted (lower panel). The data were presented as the mean ± SD. *$P$ < 0.05 compared with the indicated groups; $t$ test. NS, not significant.

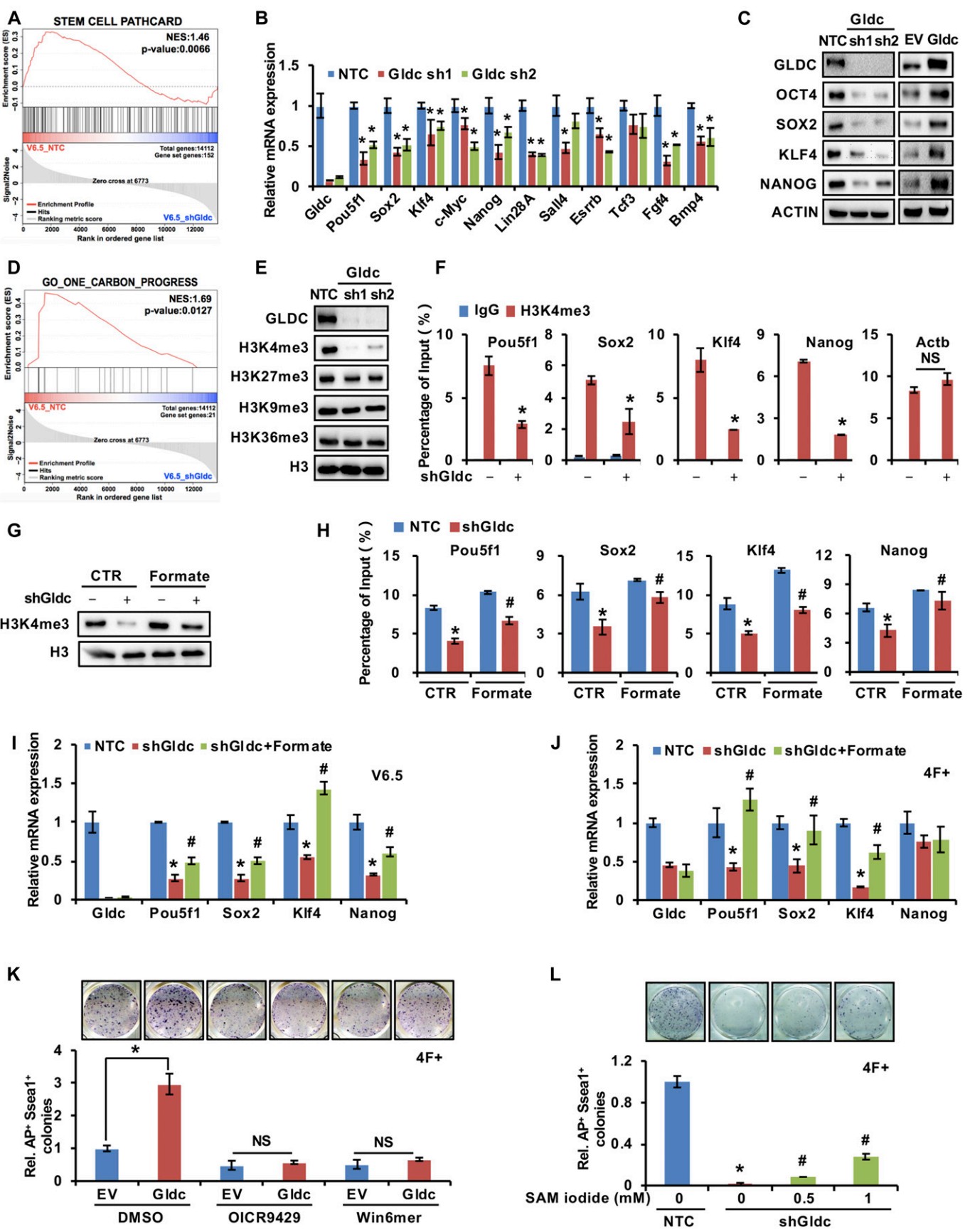

MG, as indicated by an increase in the level of argpyrimidine, leading to senescence and compromised pluripotency (Fig 6G–K). More interestingly, supplementation with SAM iodide or over-expression of GLO1 in IMR90 cells partially but significantly rescued the reprogramming efficiency suppressed by shGldc (Fig 6L). Together with our results in mouse cells, these data suggest that the GCS is activated in both mouse and human ESCs and is critical for the maintenance and acquisition of stem cell pluripotency.

## Discussion

PSCs exhibit a unique metabolic phenotype that distinguishes them from differentiated cells. Although emerging studies have focused on the specific metabolic pathways in PSCs, our understanding is still very limited in this regard. Here, we report that the GCS functions as an essential regulator of stem cell fate determination. Using a screening approach, we found that the GCS is activated in PSCs via dramatically induced expression of Gldc. Intriguingly, the activated GCS was further revealed to regulate the pluripotency of PSCs by maintaining the epigenetic state of PSCs and protecting PSCs from MG-driven senescence (Fig 7). Together, our findings establish the GCS as a previously unappreciated mechanism that is critical for the fate of PSCs.

Serine/glycine metabolism has emerged as a key metabolic node in proliferating cells such as cancer cells. Consumption and release (CORE) profiles of 219 metabolites from culture medium across the NCI-60 cancer cell lines identified glycine as the metabolite that correlated most to the rates of cell proliferation across these cancer cell lines (Jain et al, 2012). Amplification of Phgdh has been found in breast cancer (Possemato et al, 2011) and melanoma (Mullarky et al, 2011). Gldc was also found to be up-regulated in non-small-cell lung carcinoma (Zhang et al, 2012b) and glioma (Kim et al, 2015). Although PSCs are recognized to share many properties with cancer cells, little is known about serine/glycine metabolism in PSCs, and there is still little information documenting how serine/glycine metabolism might be involved in the reprogramming process. Here, we confirmed that PSCs exhibited high glycine flux, which was followed by GCS activation to catabolize glycine, thus preventing glycine accumulation while maintaining the epigenetic state. Intriguingly, we also found that the activation of GCS by Gldc has a similar effect during reprogramming. Gldc knockdown leads to a nearly blockade of reprogramming without cell death, establishing GCS activation as an essential mechanism for successful somatic cell reprogramming. Consistent with that previously reported, mouse but not human ESCs catabolize threonine to supply glycine by activating Tdh (Wang et al, 2009; Shyh-Chang et al, 2013), we found that the level of Tdh was significantly increased in mouse ESCs. However, in human ESCs, the expression of SSP enzymes was increased dramatically, compared with the moderate change in mouse ESCs (Figs 1F and 6B). This difference might explain how human ESCs acquire an advantageous metabolic state without Tdh. However, it is interesting to note that GCS activation is conserved in both mouse and human PCSs. Thus, although SSP and threonine catabolism are activated in human and mouse ESCs, respectively, to maintain glycine flux, both of these cell types rely on the GCS for further glycine catabolism and cleavage, underlining the importance of the GCS system in development.

Our discovery that GCS activation is critical for protecting PSCs from senescence is potentially significant. As irreversible arrest during the G1 transition of the cell cycle, senescence normally occurs in response to stresses such as DNA damage, oxidative stress, toxic metabolites, or aberrant oncogene expression (Campisi & d'Adda di Fagagna, 2007; Collado et al, 2007; He & Sharpless, 2017). Stem cell ageing and exhaustion, defined as a decline in stem cell numbers and function, is a hallmark of organismal ageing (López-Otín et al, 2013). Lifelong persistent stem cells are especially sensitive to the accumulation of cellular damage or toxic metabolites, which ultimately leads to the loss of stemness and results in senescence (Shyh-Chang et al, 2013; Oh et al, 2014). It has also been confirmed that senescence is the greatest barrier limiting the efficiency of successful reprogramming (Banito et al, 2009; Li et al, 2009; Utikal et al, 2009). Here, we observed that deactivation of the GCS by Gldc knockdown caused the accumulation of glycine, which is subsequently converted to MG, inducing senescence of PSCs or reprogramming cells. Notably, the brain is one of the tissues that

---

**Figure 4. The GCS is required for the maintenance of PSC pluripotency by facilitating H3K4me3.**
**(A)** GSEA of a gene set comprising stem cell pluripotency genes in shGldc- versus NTC-transfected V6.5 cells. **(B, C)** qRT–PCR (B) and Western blot (C) analysis of the regulation of pluripotency-related genes by Gldc in V6.5 cells. ACTIN served as the loading control. The data were presented as the mean ± SD of three independent experiments. *$P < 0.05$ compared with the NTC; $t$ test. **(D)** GSEA of a gene set comprising genes related to one-carbon metabolism in shGldc- versus NTC-transfected V6.5 cells. **(E)** Western blot analysis of various H3 lysine methylation in V6.5 cells expressing shGldc or the NTC. Total H3 was used as the loading control. **(F)** ChIP-qPCR analysis of H3K4me3 enrichment in the promoters of Pou5f1, Sox2, Klf4, and Nanog in V6.5 cells expressing shGldc or the NTC. Actb served as a negative control. The data were presented as the mean ± SD of three independent experiments relative to the input. *$P < 0.05$ compared with the NTC; $t$ test. **(G)** Western blot analysis of H3K4me3 modifications in V6.5 cells stably expressing shGldc or the NTC with the supplementation of 1 mM formate for 48 h. Total H3 was used as the loading control. **(H)** ChIP-qPCR analysis of H3K4me3 enrichment in the promoters of Pou5f1, Sox2, Klf4, and Nanog in V6.5 cells expressing shGldc or NTC with the supplementation of 1 mM formate for 48 h. The data were presented as the mean ± SD of three independent experiments relative to the input. *$P < 0.05$ compared with the NTC; $t$ test. **(I)** qRT-PCR analysis of the expression of Pou5f1, Sox2, Klf4, and Nanog in V6.5 cells stably expressing shGldc or the NTC with the supplementation of 1 mM formate for 48 h. The data were presented as the mean ± SD of three independent experiments. *$P < 0.05$ compared with the NTC, #$P < 0.05$ compared with the Gldc knockdown group; $t$ test. **(J)** qRT-PCR analysis of the expression of endogenous Pou5f1, Sox2, Klf4, and Nanog in reprogramming cells expressing shGldc or the NTC with the supplementation of 1 mM formate for 48 h. The data were presented as the mean ± SD of three independent experiments. *$P < 0.05$ compared with the indicated group, #$P < 0.05$ compared with the Gldc knockdown group; $t$ test. **(K)** OICR-9429, Win6mer or vehicle (DMSO) was added to the medium of Gldc-overexpressing MEF cells starting from 2 d after infection with virus expressing the four factors. AP staining (upper panel) showed the iPSC colonies formed. The AP- and Ssea1-positive iPSC colonies were counted (lower panel). The data were presented as the mean ± SD. *$P < 0.05$ compared with the indicated group; $t$ test. **(L)** SAM iodide was added to the medium of Gldc knockdown MEF cells starting from 2 d after infection with virus expressing the four factors. AP staining (upper panel) showed the iPSC colonies formed. The AP- and Ssea1-positive iPSC colonies were counted (lower panel). The data were presented as the mean ± SD. *$P < 0.05$ compared with the NTC, #$P < 0.05$ compared with the Gldc knockdown group; $t$ test. NS, not significant.
Source data are available for this figure.

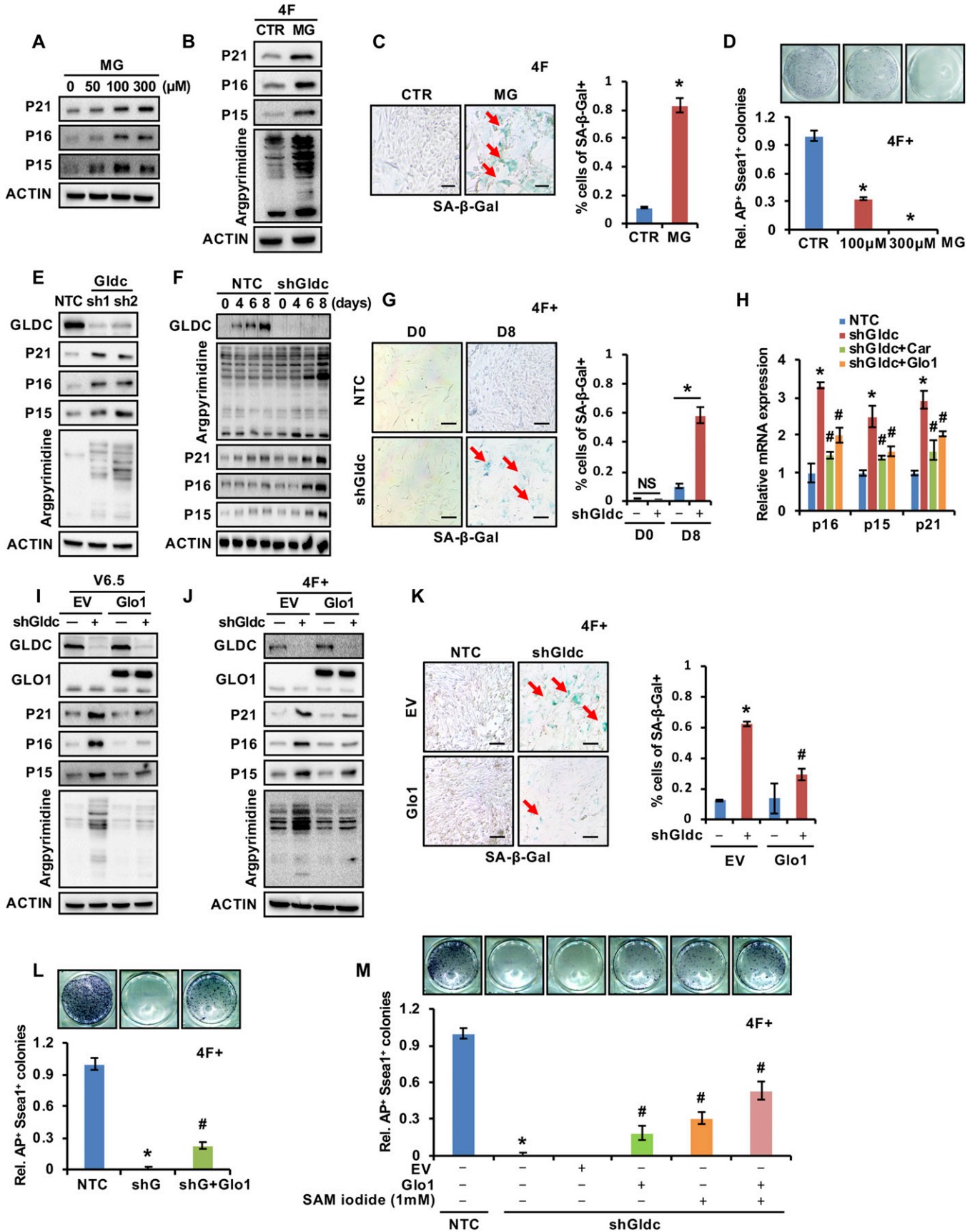

maintain high expression of Gldc (Kure et al, 2001). It was reported that loss-of-function mutations in GCS-related genes caused elevated glycine levels in body fluids, especially in the brain, a condition known as NKH. AGE accumulation in both intra- and extracellular environments has been found in the ageing brain, and MG was also proven to be involved in the progression of Alzheimer's disease and Parkinson's disease (Krautwald & Munch, 2010; Li et al, 2012; Chaudhuri et al, 2018). From our findings, it seems possible that brain ageing and age-associated neurological disorders could be related to GCS dysfunction, thus providing novel insight for understanding the pathogenesis of neurodegenerative diseases and ageing.

It has been widely accepted that the pluripotency of PSCs is related to their unusual chromatin structure or to specific histone modifications (Shyh-Chang & Ng et al, 2017). H3K4 trimethylation is associated with open euchromatin, which is crucial for the epigenetic plasticity and pluripotency of PSCs (Panopoulos et al, 2012; Denissov et al, 2014; Kidder et al, 2014). Intriguingly, we found that both mouse and human PSCs control the synthesis of SAM through activating the GCS, thereby modulating H3K4 trimethylation, which helps to maintain a pluripotent state as well as facilitates the reprogramming process. In previous studies, two groups independently proved that threonine and methionine metabolism were critical for histone methylation in mouse and human PSCs, respectively (Shyh-Chang et al, 2013; Shiraki et al, 2014). Importantly, we show here that GCS activation is conserved in both human and mouse PSCs to sustain H3K4me3 methylation and thereby control pluripotency and cell fate determination.

Although metabolism is closely correlated with the determination of stem cell fate, its significance and underlying mechanisms are largely unexplored. In this regard, it is very interesting for us to discover here that GCS activation controlled by Gldc expression in PSCs plays a vital role in pluripotency acquisition and protects stem cells from senescence, positioning Gldc as a potential target in future studies for the clinical application of stem cells. Notably, replenishing SAM and Glo1, an MG detoxifier,

only partially restored the reprogramming efficiency under Gldc knockdown conditions, suggesting that Gldc might have other potential functions beyond those described here and that warrants further investigation. Collectively, it is fundamental for us to demonstrate here that GCS activation in PSCs facilitated by Gldc expression is critical for pluripotency via epigenetic regulation and senescence control.

# Materials and Methods

### Cell culture

Feeder-independent mouse ES and iPS cells were cultured on gelatinized dishes in DMEM supplemented with 15% KnockOut Serum Replacement (Thermo Fisher Scientific), 1% FBS (Gibco), 2 mM L-GlutaMAX, 55 $\mu$M $\beta$-mercaptoethanol, 0.1 mM nonessential amino acids, 1 mM sodium pyruvate, 100 $\mu$g/ml penicillin and streptomycin, and 1,000 U/ml mouse leukaemia inhibitory factor (Millipore). Mouse ES and iPS cells cultured on feeder cells used the same DMEM as the feeder-independent mouse ES and iPS cells, except this medium contained 15% FBS (Gibco) instead of KnockOut Serum Replacement. H9 cells were cultured using TeSR-E8 (Stem Cell). MEFs were obtained from E13.5 CF-1 mouse embryos. MEF cells and IMR90 cells were cultured in DMEM supplemented with 10% FBS (Gibco), 0.1 mM nonessential amino acids, and 100 $\mu$g/ml penicillin and streptomycin. Formic acid (Formate, F0507; Sigma-Aldrich), S-(5'-adenosyl)-L-methionine iodide (A4377; Sigma-Aldrich), and L-carnosine (C9625; Sigma-Aldrich) were added to cell culture medium as indicated. V6.5 cells were kindly provided by Yangming Wang (Peking-Tsinghua Center for Life Sciences). H9 and iPS cell lines were obtained from the American Type Culture Collection. V6.5, H9, and iPS cell lines were used within 15 passages (less than 2 mo) after reviving from the frozen stocks. MEFs were used with four passages after reviving from the frozen stocks.

---

**Figure 5. The GCS protects PSCs from MG-induced senescence.**
**(A)** Western blot analysis of the expression of senescence-related genes in V6.5 cells treated with various concentration gradients of MG. ACTIN served as the loading control. **(B)** Western blot analysis of the expression of senescence-related genes in reprogramming cells treated with MG. ACTIN served as the loading control. **(C)** Reprogramming cells were treated with MG or control, and the SA-$\beta$-gal-positive cells were counted. The data were presented as the mean ± SD. *$P$ < 0.05 compared with CTR; $t$ test. Scale bars, 100 $\mu$m. **(D)** MG was added to the medium of MEF cells starting from 2 d after infection with virus expressing the four factors. AP staining (upper panel) showed the iPSC colonies formed. The AP- and Ssea1-positive iPSC colonies were counted (lower panel). The data were presented as the mean ± SD. *$P$ < 0.05 compared with CTR; $t$ test. **(E)** Western blot analysis of the expression of senescence-related genes and argpyrimidine in V6.5 cells stably expressing shGldc or the NTC. ACTIN served as the loading control. **(F)** Western blot analysis of the expression of senescence-related genes and argpyrimidine in reprogramming cells expressing shGldc the NTC at various time points. ACTIN served as the loading control. **(G)** Reprogramming cells were infected with viruses expressing shGldc or the NTC, and SA-$\beta$-gal–positive cells were counted. The data were presented as the mean ± SD. *$P$ < 0.05 compared with the indicated group; $t$ test. Scale bars, 100 $\mu$m. **(H)** V6.5 cells stably expressing shGldc or the NTC were further infected with viruses expressing pSIN-3×flag-Glo1 or were supplemented with 1 mM carnosine, followed by qRT-PCR analysis of senescence-related genes. The data were presented as the mean ± SD of three independent experiments. *$P$ < 0.05 compared with the NTC, #$P$ < 0.05 compared with the Gldc knockdown group; $t$ test. **(I)** V6.5 cells stably expressing shGldc or the NTC were further infected with viruses expressing pSIN-3×flag-Glo1 or EV, followed by Western blot analysis of senescence-related genes and argpyrimidine modification. ACTIN served as the loading control. **(J)** Reprogramming cells expressing shGldc or the NTC were further infected with viruses expressing pSIN-3×flag-Glo1 or EV, followed by Western blot analysis of senescence-related genes and argpyrimidine modification. ACTIN served as the loading control. **(K)** Reprogramming cells expressing shGldc or the NTC were further infected with viruses expressing pSIN-3×flag-Glo1 or EV, and SA-$\beta$-gal–positive cells were counted. The data were presented as the mean ± SD. *$P$ < 0.05 compared with the NTC, #$P$ < 0.05 compared with the Gldc knockdown group; $t$ test. Scale bars, 100 $\mu$m. **(L)** Reprogramming cells expressing shGldc or the NTC were further infected with viruses expressing pSIN-3×flag-Glo1. AP staining (upper panel) showed the iPSC colonies formed. The AP- and Ssea1-positive iPSC colonies were counted (lower panel). The data were presented as the mean ± SD. *$P$ < 0.05 compared with the NTC, #$P$ < 0.05 compared with the Gldc knockdown group; $t$ test. **(M)** MEFs were induced with the indicated factors. AP staining (upper panel) showed the iPSC colonies formed. The AP- and Ssea1-positive iPSC colonies were counted (lower panel). The data were presented as the mean ± SD. *$P$ < 0.05 compared with the NTC, #$P$ < 0.05 compared with the Gldc knockdown group; $t$ test. NS, not significant.
Source data are available for this figure.

## qRT-PCR

Total RNA was isolated using TRIzol. One microgram of total RNA was used to synthesize cDNA using HiScript II 1st Strand cDNA Synthesis Kit (Vazyme). qRT-PCR was performed using SYBR Green Master Mix (Vazyme) on a Bio-Rad iCycler. The primer sequences used are shown in the Table S1. The fold change of the target mRNA expression was calculated based on the cycle threshold (Ct) value, where $\Delta Ct = Ct_{target} - Ct_{18S}$ and $\Delta(\Delta Ct) = \Delta Ct_{control} - \Delta Ct_{indicated\ condition}$. The expression of all samples was normalized to 18S ribosomal RNA expression.

## Western blot

Cells were harvested and lysed using RIPA buffer (50 mM Tris–HCl, pH 8.0; 150 mM NaCl; 5 mM EDTA; 0.1% SDS; and 1% Nonidet P-40 [NP-40]) supplemented with protease inhibitor cocktail. Equal amounts of protein were loaded and separated by SDS–PAGE. Primary antibodies against the following proteins were used: OCT4 (from Stemgent); SOX2 (from Millipore); MYC (from Epitomics); KLF4, SSEA1, SHMT1, SHMT2, TDH, AMT, H3, P15, and MG (from Abcam); H3K4me3, H3K9me3, H3K27me3, and H3K36me3 (from Cell Signaling Technology); FLAG (from Sigma-Aldrich); ACTIN, phosphoglycerate dehydrogenase, PSAT1, PSPH, GCAT, GLDC, GCSH, P16, P21, and GLO1 (from Proteintech). Signals were detected using Western ECL Substrate (Bio-Rad). Antibody information is listed in Table S2.

## Induction of PSCs

For mouse iPS induction, pMX-based vectors containing mouse Oct4, Sox2, Klf4, or Myc (Addgene) or other vectors containing genes relevant to the experiment were transfected into HEK293T cells using Lipofectamine 2000 (Invitrogen). 48 h after transfection, medium containing virus particles was collected and used to infect preseeded MEF cells supplemented with 8 μg/ml polybrene (Sigma-Aldrich). 2 d after infection, the medium was replaced with fresh mouse ESC medium, which was exchanged daily with fresh medium. 6 d after infection, reprogramming cells were replated onto feeder cells. AP- or SSEA1-positive colonies were counted on day 16 after infection.

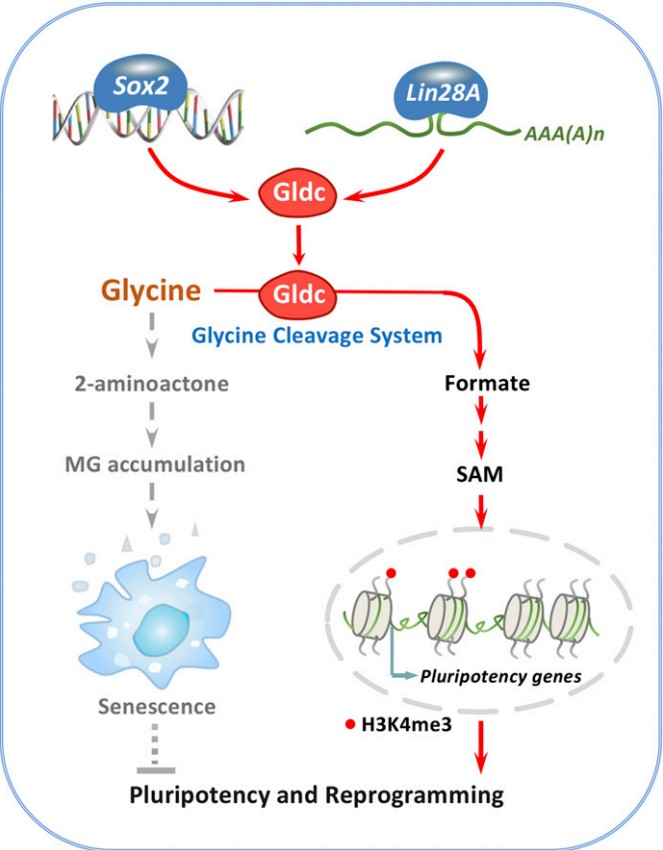

**Figure 7. Working model: The GCS determines the fate of PSCs via regulation of senescence and epigenetic modifications.**
Metabolic remodelling has emerged as critical for stem cell pluripotency; here, we provide novel evidence to demonstrate that via regulation of senescence and epigenetic modifications, the GCS determines the fate of PSCs and the outcome of the cellular reprogramming.

For human iPSC induction, IMR90 cells were seeded at a density of 100,000 cells per well in a six-well plate. Viral supernatants of transfected HEK293T cells were used for infection. Vitamin C (50 μg/ml) and valproic acid (1 mM) were added to increase the reprogramming efficiency. 8 d after infection, reprogramming cells were replated onto feeder cells, the medium was replaced daily with fresh human ESC medium, and AP-positive colonies were counted on day 35 after infection.

**Figure 6. The GCS promotes stem cell stemness and prevents cellular senescence in human stem cells.**
**(A, B)** qRT-PCR (A) and Western blot (B) analyse the expression of the enzymes in the glycine, serine, and threonine metabolic pathway in IMR90 cells and hESCs. ACTIN served as the loading control. The data were presented as the mean ± SD of three independent experiments. **(C)** The enzymatic activity of the GCS was measured in IMR90 cells and hESCs. The data were presented as the mean ± SD of three independent experiments. *$P < 0.05$ compared with IMR90 cells; $t$ test. **(D)** Western blot analysis of GLDC expression during reprogramming at the indicated time points. ACTIN served as the loading control. **(E, F)** Western blot analysis of GLDC expression in H9 cells stably expressing shSOX2 (E) or shLIN28A (F). ACTIN served as the loading control. **(G)** Western blot analysis of various H3 lysine methylation modifications in H9 cells stably expressing shGLDC or the NTC. Total H3 was used as the loading control. **(H)** Western blot analysis of argpyrimidine modifications in H9 cells stably expressing shGLDC or the NTC. ACTIN served as the loading control. **(I)** Western blot analysis of the expression of senescence-related genes in human-reprogrammed cells treated with MG. ACTIN served as the loading control. **(J)** Western blot analysis of the expression of senescence- or pluripotency-related genes in H9 cells stably expressing shGLDC or the NTC. ACTIN served as the loading control. **(K)** IMR90 cells with Gldc knockdown were infected with virus expressing the four factors to induce iPSC formation. AP staining (upper panel) showed the iPSC colonies formed. The AP-positive iPSC colonies were counted (lower panel). The data were presented as the mean ± SD. *$P < 0.05$ compared with the NTC; $t$ test. **(L)** IMR90 cells were reprogrammed with the indicated factors. AP staining (upper panel) showed the iPSC colonies formed. The AP-positive iPSC colonies were counted (lower panel). The data were presented as the mean ± SD. *$P < 0.05$ compared with the NTC, #$P < 0.05$ compared with the Gldc knockdown group; $t$ test.
Source data are available for this figure.

## Plasmids and stable cells

shRNA sequences targeting Gldc, Gcsh, Amt, Sox2, and Lin28A cloned into the pLKO (plasmid pLKO.1) vector were commercially purchased (Sigma-Aldrich). shRNA targeting sequences are listed in Table S3. The coding sequences of Gldc, Gcsh, Amt, Glo1, Oct4, Sox2, Klf4, c-Myc, Nanog, and Lin28A were subcloned into the pSIN-3×flag lentiviral vector.

## Enzymatic assay

GCS activity was measured using [$^{14}$C] glycine as previously described (Pai et al, 2015). A total of $10^7$ cells were harvested and washed once with phosphate-buffered saline (pH 7.4). The cell pellets were stored at –80°C until analysis. Then, the cell pellets were homogenized on ice in a cofactor solution containing 0.1 M Tris–Cl (pH 8.0), pyridoxal phosphate (0.4 mg/ml), dithiothreitol (1 mM), NAD (3 mg/ml), THF (5 mg/ml), and 5 $\mu$l [$^{14}$C] glycine (0.1 mCi/ ml, PerkinElmer). The samples were incubated for 2.5 h at 37°C in sealed plastic tubes with a piece of NaOH-soaked filter paper suspended inside the lid. The reaction was terminated by addition of 30% trichloroacetic acid, and the amount of $^{14}CO_2$ trapped on the filter paper was measured in a scintillation counter. Total protein from the samples was measured for quantification.

## RIP

Cells were harvested and washed with cold PBS. Cells were obtained by centrifugation and lysed using RIP buffer (50 mM Tris–HCl, pH 7.4; 0.25 M NaCl; 1% Nonidet P-40 [NP-40]; 10 mM EDTA; and RNase inhibitor [Promega]) for 45 min. Cell lysates were precleared with beads for 1.5 h; the beads were then removed, and the supernatant was incubated with antibody for 14–18 h at 4°C, followed by incubation with beads for another 2 h. Protein–RNA complexes bound to the beads were washed five times in RIP buffer. RNA extraction and cDNA synthesis were performed as described in the qRT-PCR protocol. The oligos used are listed in Table S4.

## Teratoma formation

iPSCs grown on feeder cells were harvested and suspended at 1 × $10^7$ cells/ml in DMEM containing 50% Matrigel (BD). A total of 1 × $10^6$ cells were injected into the dorsal flank of nude mice (SJA Laboratory Animal Company). After 4–8 wk, teratomas were removed and fixed in 10% formaldehyde. The samples were then embedded in paraffin, and sections were stained with haematoxylin and eosin.

## Dual luciferase reporter assay

To assess Lin28A binding, different fragments or the full-length sequence of Gldc were cloned into the pSI-check2 vector (Promega). 200 ng of luciferase reporter plasmids were transfected with either pSIN-3×flag-Lin28A or EV into HEK293 cells in each well of a 48-well plate. Firefly luciferase activity and Renilla luciferase activity were measured 48 h after transfection using a Dual Luciferase Reporter Assay System (Promega).

To assess Sox2 binding, the Gldc promoter sequence with the predicted Sox2-binding site or a mutated sequence was inserted into the pGL2-promoter luciferase reporter vector (Promega). HEK293 cells preseeded in a 48-well plate were cotransfected with 200 ng of the luciferase reporter plasmid, pSIN-3×flag-Sox2 or EV, and 4 ng of the pSV-Renilla plasmid. Firefly luciferase activity and Renilla luciferase activity were measured 48 h after transfection using the Dual Luciferase Reporter Assay System (Promega).

## MG measurement

MG levels were measured using an Methylglyoxal assay kit (ab241006; Abcam) according to the manufacturer's instructions.

## RNA-seq analysis

Total RNA was extracted using TRIzol reagent (Life Technologies) following the manufacturer's instructions, and the RNA integrity number was determined on an Agilent Bioanalyzer 2100 to assess RNA integrity. RNA-seq was performed by BGI using the Illumina HiSeq X10 platform. Reads were first mapped to the mouse reference genome mm10 using TopHat2 (v2.1.0). Transcripts were assembled by StringTie (v1.3.4 days), and gene expression analysis was performed using Ballgown v2.2.0. The accession number for RNA-seq data is Gene Expression Omnibus: GSE137439.

## Gas chromatography–mass spectrometry

Cells were collected in 80% (vol/vol) methanol, followed by repeated freezing and thawing. Insoluble material in the lysates was removed by centrifugation, and the supernatants were dried in a rotary evaporator and resuspended in 75 $\mu$l of pyridine. Metabolites were further derivatized by the addition of 25 $\mu$l of MTBSTFA containing 1% tert-butyl dimethyl chlorosilane at 60°C for 1 h. The samples were analysed using an Agilent DB-5MS column in an Agilent 7890/5975C GC/MS system (Agilent Technologies). Peaks representing each defined intensity were identified from known standards.

Formate was analysed by GC-MS as described previously (Lamarre et al, 2014). Briefly, cells were collected, and metabolites were extracted with ice-cold 80% (vol/vol) methanol, followed by centrifugation to remove insoluble material. The supernatants were dried in a rotary evaporator and resuspended in 50 $\mu$l of $H_2O$, and alkylation was carried out by mixing 20 $\mu$l of phosphate buffer with 130 $\mu$l of 100 mM pentafluorobenzyl bromide, vortexing for 1 min, and incubating the samples at 60°C for 15 min. Next, the tubes were cooled to room temperature, and 330 $\mu$l of n-hexane was added and vortexed for 1 min until the two layers were separated. A 200 $\mu$l volume of the top layer was transferred to glass inserts and analysed on a GC-MS system equipped with an Agilent HP-INNOWax GC column.

## Liquid chromatography–mass spectrometry

Cells were collected and immediately flash-frozen in liquid nitrogen. Metabolites were extracted with 80% (vol/vol) ice-cold methanol by ultrasonication, followed by centrifugation to remove insoluble material. The supernatants were evaporated and resuspended in 50 $\mu$l of 80% methanol. Samples were analysed using a 5600 Plus

TripleTOF mass spectrometer (AB/SCIEX) coupled to an HPLC system (AB/SCIEX). SAM and 5mTHF were separated on a Polar C18 HPLC column (phenomenex); AMP and GSH were separated on an HILIC HPLC column (phenomenex). Peaks representing each metabolite were extracted and analysed using MultiQuant software.

### Chromatin immunoprecipitation assay

Cells were fixed with 1% formaldehyde for 10 min and quenched in 0.125 M glycine for 5 min at room temperature. The cells were harvested and sonicated by a Scientz 92-IIN Sonication System. DNA was immunoprecipitated by either control IgG or anti-H3K4me3 primary antibody (Cell Signaling Technology) overnight at 4°C. RNA and protein were digested using RNase A and proteinase K. DNA was purified on a Sangon column, followed by qRT-PCR analysis using SYBR Green Master Mix (Vazyme). The oligos used are listed in Table S5.

### Statistical analysis

Data were presented as the mean ± SD of at least three independent experiments. *P* values were calculated by using *t* test. Statistical significance was defined as $P < 0.05$.

## Supplementary Information

## Acknowledgements

Our work was supported in part by the National Key R&D Program of China (2018YFA0800300, 2018YFA0107103, and 2017YFA0205600), the National Natural Science Foundation of China (81525022, 31571472, 81530076, and 81821001), the Strategic Priority Research Program of the Chinese Academy of Sciences (XDPB10), the Major/Innovative Program of Development Foundation of Hefei Center for Physical Science and Technology (2017FXZY004), and the Program for Guangdong Introducing Innovative and Entrepreneurial Teams (2017ZT07S054).

### Author Contribution

S Tian: data curation, formal analysis, validation, investigation, and writing—original draft, review, and editing.
J Feng: data curation, formal analysis, validation, investigation, and writing—original draft, review, and editing.
Y Cao: validation and investigation.
S Shen: validation and investigation.
Y Cai: validation and investigation.
D Yang: validation and investigation.
R Yan: validation and investigation.
L Wang: validation and investigation.
H Zhang: conceptualization and funding acquisition.
X Zhong: supervision, funding acquisition, validation, investigation, and writing—original draft, review, and editing.
P Gao: conceptualization, supervision, funding acquisition, and writing—original draft, review, and editing.

### Conflict of Interest Statement

The authors declare that they have no conflict of interest.

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
