## [Reviewer comments · Life Science Alliance]

Life Science Alliance

Glycine Cleavage System Determines the Pluripotency via Senescence and Epigenetic Regulation

Shengya Tian, Junru Feng, Yang Cao, Shengqi Shen, Yongping Cai, Dongdong Yang, Ronghui Yan, Lihua Wang, Huafeng Zhang, Xiuying Zhong, and Ping Gao

DOI: <https://doi.org/10.26508/lsa.201900413>

Corresponding author(s): Ping Gao, University of Science and Technology of China and Xiuying Zhong, South China University of Technology

Review Timeline:

Submission Date:	2019-05-03
Editorial Decision:	2019-06-05
Revision Received:	2019-08-22
Editorial Decision:	2019-09-11
Revision Received:	2019-09-15
Accepted:	2019-09-17

Scientific Editor: Andrea Leibfried

Transaction Report:

June 5, 2019

Re: Life Science Alliance manuscript #LSA-2019-00413-T

Prof. Ping Gao
University of Science and Technology of China
School of Life Sciences
Hefei, Anhui 230026
China

Dear Dr. Gao,

Thank you for submitting your manuscript entitled "Glycine Cleavage System Determines the Pluripotency via Senescence and Epigenetic Regulation" to Life Science Alliance. The manuscript was assessed by expert reviewers, whose comments are appended to this letter.

As you will see, the reviewers appreciate your findings but think that some of your conclusions need better support. Based on the reviewer input, we would like to invite you to submit a revised manuscript to us. Importantly, the requested controls, quantifications and clarifications as well as the requested better support for the conclusions regarding pluripotency/differentiation as well as senescence should get provided. Reviewer #2 also asks for a few experiments to better support your claims and to provide more detailed insight, and we would like to ask you to perform these. We do not expect you to solve the noted discrepancy to the related work by Kang et al experimentally, but the discrepancy should get discussed. We realize that addressing all these concerns, and particularly those of reviewer #1, is feasible but demanding. We think that the concerns are however justified and aim at strengthening your study. We can extend the standard revision time, should this be helpful.

Thank you for this interesting contribution to Life Science Alliance. We are looking forward to receiving your revised manuscript.

Sincerely,

B. MANUSCRIPT ORGANIZATION AND FORMATTING:

Reviewer #1 (Comments to the Authors (Required)):

Summary/Description of Advancement in the Field:

This submission identifies a specific pathway of amino acid metabolism - the glycine cleavage system - as an important regulator of pluripotency. Authors find that an important enzyme in the glycine cleavage system, GLDC, is regulated by Sox2 and LIN28A. In turn, GLDC regulates pluripotency through production of one-carbon units that influence H3K4me3 deposition, and maintains pluripotency through prevention of methylglyoxal (MG)-induced senescence. This work elucidates a mechanism by which GLDC is controlled, and provides evidence for how the glycine cleavage system regulates pluripotency exit in mice and humans.

Figure 1: Authors claim that the glycine cleavage system in mice is highly activated in pluripotent cells due to high levels of GLDC, contrary to their fibroblast counterparts. They claim that the activity of the GLDC-mediated glycine cleavage system is associated with pluripotency. Authors identify high levels of glycine metabolism (RNA-Seq, western blot) in pluripotent cells compared to fibroblasts, and characterize an associated increased expression of enzyme GLDC in various pluripotent states.

Figure 1 Comments:

Figure 1J

a) This figure conflicts with what the main text is stating, which is that iPSCs and mESCs consume higher levels of glycine because of high activity of the glycine cleavage system due to increased levels of GLDC. However, this Y-axis label (extracellular glycine consumption) implies that iPSCs and mESCs exhibit lower levels of glycine consumption compared to MEFs. The authors should clarify this.

b) The figure legend indicates all cells were cultured first in glycine deprivation media for 12 hours, followed by treatment with labeled glycine to measure glycine consumption. How does this initial glycine deprivation affect viability and/or subsequent glycine consumption due to compensatory uptake? Could glycine deprivation induce differentiation, and mESCs and iPSCs being measured may no longer be a homogenous population of pluripotent cells?

Figure S1B/1C/1D

a) Authors should verify/show that reprogrammed cells express pluripotent markers on this western blot figure.

Figure 2: Authors find that Sox2 and Lin28A regulate GLDC expression. Sox2 regulates GLDC expression through transcriptional binding of the GLDC promoter. In contrast, LIN28A regulates GLDC expression through post-transcriptional means by binding GLDC mRNA. Double knockdown of both Sox2 and Lin28A causes significant decrease in GLDC expression.

Figure 3: Authors evaluate regulation of metabolite levels by GLDC, and find that GLDC is important for preventing methylglyoxal accumulation. Using metabolic flux assays, they further determine that GLDC is also essential for one-unit carbon production of SAM and formate. Formate production appears to rescue pluripotency whereas MG supplementation promotes senescence. Formate production and methylglyoxal levels are determined to help regulate pluripotency.

Figure 3 Comments:

Figure 3J

a) GLDC knockdown correlates with decreased pluripotency as shown by AP+ colonies and immunofluorescence, is there a corresponding increase in differentiation markers?

Figure 3L

a) SAM levels are important substrates for deposition of H3K4me3, which help maintain pluripotency. Are even more reprogrammed pluripotent colonies induced when supplemented with formate and SAM? Why do authors select formate to rescue the pluripotent phenotype in 4F+ knockdowns?

Figure 4: Authors claim that GLDC mediates pluripotency regulation through one carbon-unit production that contributes to H3K4me3 occupancy at transcription start sites of key pluripotent genes.

Figure 4 Comments:

Figure 4G

a) How does formate restore H3K4me3 levels? In other words, how is formate a methyl donor in this instance? Are authors using methyl formate to restore H3K4me3?

b) Formate is a negative ion, likely discouraging cell permeability. Authors should confirm that formate is entering the cell in these experiments when rescuing GLDC knockdowns.

Figure 5: Authors claim that the glycine cleavage system in pluripotent cells prevents MG-induced senescence.

Figure 5 Comments:

Figure 5A

a) Titration of MG should say 100 μ M, not 10 μ M.

Figure 5E

a) Authors should treat MEFs with MG as a negative control, confirming that high GLDC activity preventing MG accumulation is exclusive to the pluripotent state. A titration should be done to show the pluripotent state is specifically more sensitive to MG supplementation.

Figure 5M

a) Why do authors switch between formate and SAM rescues of GLDC knockdowns?

General

a) How does addition of MG to pluripotent cells/during reprogramming affect H3K4me3 marks? From Figure 3M, supplementation of MG causes decreases in AP positive colonies - is this caused by senescence or H3K4me3?

b) In pluripotent cells, is there an enrichment for MG detoxification systems, such as glyoxylase pathways, compared to differentiated cells? How do pluripotent cells respond to MG intolerance?

Figure 6: Authors demonstrate that the glycine cleavage system regulates pluripotency by preventing MG-induced senescence and controlling H3K4me3 deposition in human cells.

Figure 6 Comments:

Figure 6G

a) Decreases in both H3K4me3 and H3K27me3 are seen in GLDC knockdowns of hPSCs. Can authors comment on the decrease in H3K27me3?

General

a) Figure 6 should include confirmation that reprogrammed human cells treated with MG give higher expression of senescence markers.

Figure 7: The schematic indicates that formate is derived from SAM. Figure 3 identifies formate and SAM as two separate byproducts of the glycine cleavage system. The authors should explain this discrepancy.

Other:

- a) It has been reported that an increase in H3K4me3 is seen in GLDC knockdowns of hPSCs (Kang PJ, et al., *Metabolic Engineering* 2019; 53: 35-47), while the authors of this paper see a decrease in H3K4me3 in hPSCs with GLDC knockdowns. Authors should explain this discrepancy.
- b) Kang et al. also claims that Klf4 and c-myc regulate GLDC expression, whereas authors here claim that Sox2 and Lin28A actually regulate GLDC expression. Authors should explain this discrepancy.
- c) Methods- Identify the passage number/population doubling number of cells used for each of the experiments, as well as from where they were procured.
- d) Discussion- Stem cell aging and exhaustion is a hallmark but not necessarily the primary driver of organismal aging. Avoid generalizations and see Hallmarks of Aging: Cell paper by Carlos Lopez-Otin.
- e) Introduction/Discussion- Not all cancers are primarily glycolytic, so correct for generalized statements

Reviewer #2 (Comments to the Authors (Required)):

In this study Tian et al. discover that the glycine cleavage system (GCS) is involved in promoting pluripotency in original stem cells as well as during reprogramming. The proposed mechanisms involve a general promotion of H3K4 trimethylation as well as prevention of methylglyoxal accumulation a cytotoxin that counteracts pluripotency by inducing cellular senescence. The study is written clearly, contains several interesting insights that are important to the field and reveals a so far unacknowledged role of Gldc and the GCS in stem cell biology.

major comments:

Fig2

A

The Western Blot shows an induction of GLDC expression upon Sox2 and Lin28A expression. Moreover, it seems that also Klf4 (probably mislabeled as Kif4 in the Figure) has the same effect. Could the authors please comment on that and if possible, provide a quantification of the band intensities across several experiments in order to be able to judge the effect size?

M

The effect of Sox2 and Lin28A single knock downs on GLDC expression seems to be very mild, especially in comparison to experiments shown in 2B and 2F, again a quantification would help.

In general, this figure provides evidence of transcriptional (Sox2) and posttranscriptional (Lin28A) control of GLDC expression. However, in 1G the authors show and also explicitly state in the result section of the text that transcriptional induction of GLDC during reprogramming precedes induction of 4F including Sox2. The authors should discuss the apparent discrepancy and if possible, formulate alternative hypothesis for Sox2-independent GLDC transcriptional control.

Fig4

A and D

The representation of the GSEA needs improvement since the current figure is not readable at the presented resolution. Also, it would be helpful to display the total number of genes / gene sets analyzed and housekeeping pathways that may not change as controls.

B

The panel shows that pluripotency marker gene expression is lost upon reduced *Gldc* expression. How specific is this effect? Are markers for different germ layers induced, are there any molecular or morphological hints on into what cellular state these cells differentiate into?

C

The downregulation of *SOX2* upon *Gldc* knock down and opposite behavior upon overexpression suggests a positive feedback loop between these factors. Could the authors please comment on that possibility and how it fits in their proposed model Fig7?

F

The pluripotency marker genes tested for H3K4me3 tested in Figure 4F suggest an expression and cell identity change due to a reduction of methylation levels of about 50 % in a *Gldc* knock down. Do the authors have control data from non-pluripotent associated genes or, ideally, could they provide genome-wide ChiP-Seq data to understand how selective the proposed action of the GCS is? Is it possible that this effect is a side-effect of the proposed anti-senescence activity observed in data of Fig5? Given that conclusion and model of this work are largely based on this functional connection it deserves further investigation.

I

What is the effect of formate only on these genes? This experiment might reveal if formate is rate limiting in the proposed network and might explain the observed overshoot for *Klf4*.

Fig5D

How do the used MG concentrations compare to endogenously observed intracellular concentrations?

Minor comments

Could the authors discuss how the observed changes in *Gldc* expression levels compare to differential expression during mouse embryogenesis?

Point-by-point response and revision results

Editor's Decision:

Thank you for submitting your manuscript entitled "Glycine Cleavage System Determines the Pluripotency via Senescence and Epigenetic Regulation" to Life Science Alliance. The manuscript was assessed by expert reviewers, whose comments are appended to this letter.

As you will see, the reviewers appreciate your findings but think that some of your conclusions need better support. Based on the reviewer input, we would like to invite you to submit a revised manuscript to us. Importantly, the requested controls, quantifications and clarifications as well as the requested better support for the conclusions regarding pluripotency/differentiation as well as senescence should get provided. Reviewer #2 also asks for a few experiments to better support your claims and to provide more detailed insight, and we would like to ask you to perform these. We do not expect you to solve the noted discrepancy to the related work by Kang et al experimentally, but the discrepancy should get discussed. We realize that addressing all these concerns, and particularly those of reviewer #1, is feasible but demanding. We think that the concerns are however justified and aim at strengthening your study. We can extend the standard revision time, should this be helpful.

Response: Thank you very much for your kind decision to allow us to revise the manuscript. We appreciate the constructive comments and suggestions from the reviewers. We have carried out multiple experiments that the reviewers suggested and revised the manuscript accordingly. Please find attached our point-by-point response to the reviewers' concern. Thank you for your consideration and hope you find our responses satisfactory.

For the reviewers' convenience, we have appended in this file all the revised figures, which we labeled as **Figure R1** to **Figure R19**.

Referee #1:

This submission identifies a specific pathway of amino acid metabolism - the glycine cleavage system - as an important regulator of pluripotency. Authors find that an important enzyme in the glycine cleavage system, GLDC, is regulated by Sox2 and LIN28A. In turn, GLDC regulates pluripotency through production of one-carbon units that influence H3K4me3 deposition, and maintains pluripotency through prevention of methylglyoxal (MG)-induced senescence. This work elucidates a mechanism by which GLDC is controlled, and provides evidence for how the glycine cleavage system regulates pluripotency exit in mice and humans.

Response: We are grateful for the reviewer's comments that well summarized the major findings and significance of our work.

Figure 1: Authors claim that the glycine cleavage system in mice is highly activated in pluripotent cells due to high levels of GLDC, contrary to their fibroblast counterparts. They claim that the activity of the GLDC-mediated glycine cleavage system is associated with pluripotency. Authors identify high levels of glycine metabolism (RNA-Seq, western blot) in pluripotent cells compared to fibroblasts, and characterize an associated increased expression of enzyme GLDC in various pluripotent states.

Response: Thank you for the comments that well summarized **Figure 1** in our work.

Figure 1 Comments:

Figure 1J

a) This figure conflicts with what the main text is stating, which is that iPSCs and mESCs consume higher levels of glycine because of high activity of the glycine cleavage system due to increased levels of GLDC. However, this Y-axis label (extracellular glycine consumption) implies that iPSCs and mESCs exhibit lower levels of glycine consumption compared to MEFs. The authors should clarify this.

Response: We thank the reviewer for pointing out this mistake. We apologize for using an inappropriate Y-axis label which conveyed inaccurate information. We have corrected the Y-axis label in **Figure 1J** to "Relative levels of residual ¹³C-glycine in

medium”, which indicated that iPSCs and mESCs exhibit high levels of glycine utilization compared to MEFs. For your convenience, we have also appended this figure here as **Figure R1A**.

Figure R1. mESCs and iPSCs exhibit high levels of glycine utilization compared to MEFs. (A) MEFs, mESCs and iPSCs were first cultured in glycine starvation medium for 12 hours, and the medium was then refreshed with medium containing ^{13}C -labelled glycine. The amount of ^{13}C -labelled glycine in the culture medium from MEFs, iPSCs and mESCs was measured by GC/MS at the indicated time. The data were presented as the mean \pm SD of three independent experiments. * $P < 0.05$ compared with MEFs.

b) The figure legend indicates all cells were cultured first in glycine deprivation media for 12 hours, followed by treatment with labeled glycine to measure glycine consumption. How does this initial glycine deprivation affect viability and/or subsequent glycine consumption due to compensatory uptake? Could glycine deprivation induce differentiation, and mESCs and iPSCs being measured may no longer be a homogenous population of pluripotent cells?

Response: Glycine is in a dynamic exchange in the intracellular and extracellular space. In order to better exhibit the glycine consumption and increase the isotope labelling efficiency, we cultured the cells in glycine deprivation medium for a short term before isotope labelling. To address the reviewer’s concern, we further analyzed the glycine consumption of mESCs and MEFs with or without short-term of glycine starvation in advance, and the results confirmed that ESCs exhibit a greater glycine utilization than MEFs either with or without short-term of glycine starvation in advance (**Figure R2A**). Furthermore, we detected the viability and pluripotency of mESCs under the condition of glycine starvation. As a result, no significant difference

was detected in the viability (**Figure R2B**) or pluripotency of mESCs between that cultured under normal condition and that under short term of glycine deprivation (**Figure R2C**), which is consistent with the previous reports ¹.

Figure R2. Short-term of glycine starvation didn't influence the viability or pluripotency of mESCs. (A) MEF and V6.5 cells were first cultured under normal or glycine deprivation condition for 12 hours, and the medium was then refreshed with medium containing ¹³C-labelled glycine. The amount of ¹³C-labelled glycine in the culture medium of MEFs or V6.5 cells was measured by GC/MS 0 or 4 hours after medium change. Bar graph presented the relative levels of residual ¹³C-labelled glycine in the culture medium of the indicated cells compared to that of MEFs or V6.5 cells 0 hours after medium change. The data were presented as the mean ± SD of three independent experiments. (B) Cell numbers of V6.5 cells cultured under normal or glycine deprivation condition for 12 hours. Cell numbers were counted after trypan blue staining. Data were presented as mean ± SD. (C) qRT-PCR analysis of the expression of pluripotency-related genes in V6.5 cells cultured under normal or glycine deprivation condition for 12 hours. The data were presented as mean ± SD of three independent experiments.

Figure S1B/1C/1D

a) Authors should verify/show that reprogrammed cells express pluripotent markers on this western blot figure.

Response: We agree with the reviewer that it is important to verify the expression of pluripotent markers on these reprogrammed cells. However, at such early stages of reprogramming induction, it is very difficult to distinguish the endogenous Pou5f1, Sox2, Nanog from the exogenous ones which were forced expressed during reprogramming at protein level. Alternatively, we detected the mRNA levels of endogenous Pou5f1, Sox2, Nanog by using primers targeting the 3'UTR of these pluripotency genes (**Figure R3A**, also as **Figure 1G** in original manuscript; **Figure R3B, R3C**, also as **Supplementary Figure 1C, 1D** in the revised manuscript). As a result, during somatic reprogramming process the endogenous pluripotency genes was gradually induced, and the expression of Glc3 increased even before the induction of

endogenous pluripotency genes, suggesting the important role of *Gldc* in iPSC generation. We thank the reviewer for the important point and we have included these results in **Supplementary Figure 1C, 1D** in the revised manuscript, respectively.

Figure R3. The expression levels of pluripotent markers in multiple reprogrammed cells. (A-C) qRT-PCR analysis of the expression of *Gldc*, endogenous *Pou5f1*, *Sox2* and *Nanog* in cells during OSKM-induced reprogramming (A), OSK-induced reprogramming (B) or OSNA-induced reprogramming (C) at the indicated times. The mRNA levels were normalized to the expression levels on Day 0. * $P < 0.05$ compared with the expression levels on Day 0. (Also as Supplementary Figure 1C, 1D in the revised manuscript)

Figure 2: Authors find that Sox2 and Lin28A regulate GLDC expression. Sox2 regulates GLDC expression through transcriptional binding of the GLDC promoter. In contrast, LIN28A regulates GLDC expression through post-transcriptional means by binding GLDC mRNA. Double knockdown of both Sox2 and Lin28A causes significant decrease in GLDC expression.

Response: We appreciate the reviewer for well summarizing our results in **Figure 2**.

Figure 3: Authors evaluate regulation of metabolite levels by GLDC, and find that GLDC is important for preventing methylglyoxal accumulation. Using metabolic flux assays, they further determine that GLDC is also essential for one-unit carbon production of SAM and formate. Formate production appears to rescue pluripotency whereas MG supplementation promotes senescence. Formate production and methylglyoxal levels are determined to help regulate pluripotency.

Response: We are grateful for the reviewer's comments that well summarized our results in **Figure 3**.

Figure 3 Comments:

Figure 3J

a) *GLDC knockdown correlates with decreased pluripotency as shown by AP+ colonies and immunofluorescence, is there a corresponding increase in differentiation markers?*

Response: This is indeed an important and relevant question. To investigate whether knockdown of *Gldc* would correlate with the differentiation of ESCs, the mRNA levels of some differentiation related genes in V6.5 cells expressing sh*Gldc* were analyzed, and the results showed that genes associated with endodermal differentiation, such as *Gata3* and *Wnt2*, were upregulated in *Gldc*-knockdown cells (Figure R4A).

Figure R4. The effect of *Gldc* knockdown on the expression of differentiating genes. (A) qRT-PCR analysis of the expression of differentiating genes in V6.5 cells with *Gldc* knockdown. The data were presented as the mean \pm SD of three independent experiments. * $P < 0.05$ compared with the NTC.

Figure 3L

a) *SAM levels are important substrates for deposition of H3K4me3, which help maintain pluripotency. Are even more reprogrammed pluripotent colonies induced when supplemented with formate and SAM? Why do authors select formate to rescue the pluripotent phenotype in 4F+ knockdowns?*

Response: To address this point, we further analyzed the reprogramming efficiency of *Gldc*-knockdown MEFs induced by Yamanaka Factors and supplemented with both formate and SAM. Our results showed that supplementation with either formate or

SAM could partially rescue the shGldc-suppressed reprogramming efficiency, while supplementation with both formate and SAM did not further improve the efficiency (Figure R5A). Since formate is a major donor of one-carbon unit to produce SAM (Figure R5B)^{2,3}, it is possible that formate supplementation is sufficient to support SAM production and, as a result, additional SAM supplementation did not further improve the efficiency.

Formate serves as the major one-carbon donor for SAM production and also an important intermediate metabolite of one-carbon unit metabolism promoted by Gldc. As the reviewer has pointed out that “SAM levels are important substrates for deposition of H3K4me3”, we thus used formate, the donor of one-carbon unit to produce SAM, to rescue the efficiency of somatic reprogramming caused by Gldc-knockdown.

Figure R5. Supplementation with formate and SAM partially rescued the shGldc-suppressed reprogramming efficiency. (A) SAM iodide and formic acid were supplemented to Gldc-knockdown MEF cells starting from 2 days after infection with virus expressing the four factors. AP staining (upper panel) showed the iPSC colonies formed. The AP- and Ssea1-positive iPSC colonies were counted (lower panel). The data were presented as the mean \pm SD. (B) Schematic diagram of one-carbon metabolism.

Figure 4: Authors claim that GLDC mediates pluripotency regulation through one carbon-unit production that contributes to H3K4me3 occupancy at transcription start sites of key pluripotent genes.

Response: We thank the reviewer for the comments that well summarized our results in Figure 4.

Figure 4 Comments:

Figure 4G

a) How does formate restore H3K4me3 levels? In other words, how is formate a methyl donor in this instance? Are authors using methyl formate to restore H3K4me3?

Response: In one-carbon metabolism, formate transfers a single carbon atom to THF to generate formyl-THF (CHO-THF), and formyl-THF further transforms into 5,10-methylene-THF (CH₂-THF) that finally become the one-carbon donor for SAM synthesis (**Figure R5B**). Formate serves as a one-carbon units replenisher, and further provide one-carbon units for SAM synthesis to restore H3K4me3 levels. Formic acid (HCOOH) which is widely used to restore the level of one carbon unit in cells was used in our work^{2,3}.

b) Formate is a negative ion, likely discouraging cell permeability. Authors should confirm that formate is entering the cell in these experiments when rescuing GLDC knockdowns.

Response: Thank you for the concern. Formic acid which has been reported to restore the level of one carbon unit in the cells was used in our experiments. Please also refer to the reference^{2,3}. We apologize for not making this point clear and have revised the Methods section accordingly.

Figure 5: Authors claim that the glycine cleavage system in pluripotent cells prevents MG-induced senescence.

Response: Thank you for the comments.

Figure 5 Comments:

Figure 5A

a) Titration of MG should say 100 μ M, not 10 μ M.

Response: We apologize for the error and have corrected the typo in the revised manuscript.

Figure 5E

a) Authors should treat MEFs with MG as a negative control, confirming that high GLDC activity preventing MG accumulation is exclusive to the pluripotent state. A titration should be done to show the pluripotent state is specifically more sensitive to MG supplementation.

Response: Thank you for this important and relevant question. Following this suggestion, we treated mES cells and MEF cells with MG and tested the protein levels of senescence markers. Our results showed that treatment with 50 μ M, 100 μ M MG significantly induced the protein levels of senescence markers in V6.5 cells (**Figure R6A**, also as **Figure 5A** in original manuscript) but only had slight effect on MEF cells (**Figure R6B**). Consistently, it has been reported that high concentrations of MG leads to the appearance of senescent phenotype of human skin fibroblast in previous study⁴. These data suggested that pluripotent stem cells are more sensitive to MG compared with differentiated cells.

Figure R6. V6.5 cells were more sensitive to MG treatment compared with MEF cells. (A-B) Western blot analysis of the expression of senescence-related genes in V6.5 cells (A) or MEF cells (B) treated with various concentration gradients of MG. ACTIN served as the loading control.

Figure 5M

a) Why do authors switch between formate and SAM rescues of GLDC knockdowns?

General

Response: Thank you for the comments. Firstly, we found Glc promoted one-carbon unit metabolism, which might be associated with its effect on somatic reprogramming. Then formate, the intermediate metabolites of this pathway and also the donor of

one-carbon unit to produce SAM, was used to test whether it could rescue the efficiency of somatic reprogramming caused by Gldc-knockdown. To further test whether Gldc promotes pluripotency acquisition via facilitating the production of SAM, the donor of methylation, we used SAM to rescue Gldc-knockdowns. To address the reviewer's concern, we further tested the effects of formate and SAM on somatic reprogramming induced by Yamanaka Factors with or without Gldc-knockdown. The results showed that there is no obvious difference of rescue effect between formic acid and SAM iodide (**Figure R7A**).

Figure R7. Supplementation with formate or SAM rescued the shGldc-suppressed reprogramming efficiency equally. (A) SAM iodide and formic acid were added to the medium of Gldc knockdown MEF cells starting from 2 days after infection with virus expressing the four factors. AP staining (upper panel) showed the iPSC colonies formed. The AP- and Ssea1-positive iPSC colonies were counted (lower panel). The data were presented as the mean \pm SD.

a) How does addition of MG to pluripotent cells/during reprogramming affect H3K4me3 marks? From Figure 3M, supplementation of MG causes decreases in AP positive colonies - is this caused by senescence or H3K4me3?

Response: This is an important and relevant question. In order to address this point clearly, we detected the influence of MG on H3K4me3 in V6.5 cells. The results showed that supplementation of MG had no effect on H3K4me3 levels in mESCs. (**Figure R8A**), indicating that MG accumulation caused decreases in generation of AP positive colonies via inducing senescence, but not affecting H3K4me3 levels.

Figure R8. Supplementation of MG have no effect on H3K4me3 level. (A) Western blot analysis of H3K4me3 modifications in V6.5 cells treated with various concentration gradients of MG. Total H3 served as the loading control.

b) In pluripotent cells, is there an enrichment for MG detoxification systems, such as glyoxylase pathways, compared to differentiated cells? How do pluripotent cells respond to MG intolerance?

Response: The glyoxalase system, comprised of Glo1 and Glo2, catalyzes the conversion of methylglyoxal (MG) to D-lactate via the intermediate S-d-lactoylglutathione. Glo1 catalyzes the first step of MG detoxification, which metabolizes MG to S-d-lactoylglutathione. S-d-lactoylglutathione is then converted to D-lactate by Glo2. To address the reviewer' concern, we detected the expression levels of Glo1 and Glo2 in MEFs and ESCs. qRT-PCR and western blot analysis revealed that the expression levels of Glo1 and Glo2 were upregulated in V6.5 cells as compared to MEFs (**Figure R9A, R9B**), which is consistent with previous report⁵.

Methylglyoxal is a toxic, highly reactive aldehyde which can react with lipids, nucleic acids, or with lysine and arginine residues of proteins to form advanced glycation end products (AGEs). As we find in our study, MG treatment induced the expression of senescence markers in ESCs dose-dependently, suggesting that MG intolerance may lead to senescence of ESCs.

Figure R9. Glo1 and Glo2 were upregulated in V6.5 cells as compared to MEFs. (A-B) qRT-PCR (A) and western blot (B) analysis of the expression of Glo1, Glo2 in MEF and

V6.5 cells. The data were presented as the mean \pm SD of three independent experiments. *P<0.05 compared with MEFs. ACTIN served as the loading control.

Figure 6: Authors demonstrate that the glycine cleavage system regulates pluripotency by preventing MG-induced senescence and controlling H3K4me3 deposition in human cells.

Response: We appreciate the reviewer for well summarizing our results in **Figure 6**.

Figure 6 Comments:

Figure 6G

a) Decreases in both H3K4me3 and H3K27me3 are seen in GLDC knockdowns of hPSCs. Can authors comment on the decrease in H3K27me3?

Response: Thank you for this comment. We repeated the experiments and provided the results of three independent experiments here with quantification of the band intensities by the Image lab 3.0 software (Bio-Rad). The results showed that knockdown of Glc significantly reduced H3K4me3 while the level of H3K27me3 was slightly downregulated but not significantly (**Figure R10A**, also as **Figure 6G** in the original manuscript; **R10B**). Hence, the current study focused on the regulation of Glc on H3K4me3.

(Fig 6G in the original manuscript)

Figure R10. The effect of Glc knockdown on the level of H3K4me3 and H3K27me3 in hPSCs. (A) Western blot analysis of various H3 lysine methylation modifications in H9 cells stably expressing shGLDC or the NTC. Total H3 was used as the loading control (Also as Figure 6G in the original manuscript). (B) Western blot analysis of H3K4me3 and H3K27me3 modifications in H9 cells expressing shGLDC or the NTC. Total H3 was used as the loading control.

General

a) Figure 6 should include confirmation that reprogrammed human cells treated with MG give higher expression of senescence markers.

Response: Thank you for this important suggestion. To address this point, we treated the reprogrammed human cells induced by Yamanaka Factors with MG and detected the expression of senescence markers. The results showed that supplementation of MG increased the expression levels of senescence markers in reprogrammed human cells (**Figure R11A**). Per our reviewer's suggestion, we have included the result as **Figure 6I** in the revised manuscript.

Figure R11. MG induced senescence in human reprogrammed cells. (A) Western blot analysis of the expression of senescence-related genes in Yamanaka Factors-induced human reprogrammed cells treated with MG. ACTIN served as the loading control. (Also as Figure 6I in the revised manuscript)

Figure 7: The schematic indicates that formate is derived from SAM. Figure 3 identifies formate and SAM as two separate byproducts of the glycine cleavage system. The authors should explain this discrepancy.

Response: We apologize for providing the improper information in **Figure 3A** and have revised the schematic diagram. For your convenience, we have also appended this figure here as **Figure R12A**. In addition, a more detailed schematic diagram was provided here for better understanding one-carbon metabolism (**Figure R5B**). The direct product of the glycine cleavage system is CH_2 -THF. CH_2 -THF further turns to CHO-THF and then to formate in mitochondria. Formate is then transported to cytoplasm and provides one-carbon unit to produce CH_2 -THF and finally reacts with homocysteine (Hcy), generating methionine, the precursor of SAM.

Figure R12. Schematic diagram of metabolic flux related to the GCS. (Also as Figure 3A in the revised manuscript)

Figure R5B. Schematic diagram of one-carbon metabolism.

Other:

a) It has been reported that an increase in H3K4me3 is seen in GLDC knockdowns of hPSCs (Kang PJ, et al., Metabolic Engineering 2019; 53: 35-47), while the authors of this paper see a decrease in H3K4me3 in hPSCs with GLDC knockdowns. Authors should explain this discrepancy.

Response: Thank you for the comments, we also noticed this discrepancy. To address this concern, we have repeated our experiment. Consistently with our previous data, the results showed that knockdown of Glc decreased H3K4me3 level in human ESCs (**Figure R10A, R10B**). Then we carefully compared the methods between these two works, and consider that such discrepancy might be caused by different cell culture condition and different cell lines we used in the experiments. In detail, H9 cell line

was used in our study, while BG01 cell line was used in the study of Kang et al. The cell culture condition was also different. The classical TeSR-E8 medium (from Stem Cell) was used to culture H9 cells in our study, while DMEM/F12 supplemented with 2× ITS, 64 µg/ml L-ascorbic acid-2-phosphate, 100 ng/ml FGF2, 543 µg/ml NaHCO₃, and 2 ng/ml TGF-β1 was used to culture BG01 cells in the study of Kang et al. However, further studies are required for the exploration of this question.

(Fig 6G in the original manuscript)

Figure R10. The effect of Glc knockdown on the level of H3K4me3 and H3K27me3 in hPSCs. (A) Western blot analysis of various H3 lysine methylation modifications in H9 cells stably expressing shGLDC or the NTC. Total H3 was used as the loading control. (Also as Figure 6G in the original manuscript) (B) Western blot analysis of H3K4me3 and H3K27me3 modifications in H9 cells expressing shGLDC or the NTC. Total H3 was used as the loading control.

b) Kang et al. also claims that Klf4 and c-myc regulate GLDC expression, whereas authors here claim that Sox2 and Lin28A actually regulate GLDC expression. Authors should explain this discrepancy.

Response: we are grateful for the reviewer's comments. As showed in **Fig R13A** (also as **Figure 2A** in the original manuscript), our results demonstrated that multiple genes might affect Glc expression in varying degrees, among which Sox2 and Lin28A upregulated Glc significantly while the other genes showed marginal effects.

Figure R13. Sox2 and Lin28A cooperatively controlled Gldc expression. (A) qRT-PCR and western blot analysis of the expression of Gldc in MEF cells overexpressing Oct4, Sox2, Klf4, c-Myc, Nanog and Lin28A individually. ACTIN served as the loading control. The data were presented as the mean \pm SD of three independent experiments. * $P < 0.05$ compared with the EV control. (Also as Figure 2A in the original manuscript)

c) Methods- Identify the passage number/population doubling number of cells used for each of the experiments, as well as from where they were procured.

Response: We thank the reviewer for the helpful advice and have revised the method accordingly.

d) Discussion- Stem cell aging and exhaustion is a hallmark but not necessarily the primary driver of organismal aging. Avoid generalizations and see Hallmarks of Aging: Cell paper by Carlos Lopez-Otin.

Response: Thank you for the comments. We have revised the Discussion section in the manuscript accordingly and have included this important reference to our revised manuscript.

e) Introduction/Discussion- Not all cancers are primarily glycolytic, so correct for generalized statements

Response: Thank you for the comments. We have revised the manuscript accordingly.

Referee #2:

In this study Tian et al. discover that the glycine cleavage system (GCS) is involved in promoting pluripotency in original stem cells as well as during reprogramming. The proposed mechanisms involve a general promotion of H3K4 trimethylation as well as prevention of methylglyoxal accumulation a cytotoxin that counteracts pluripotency by inducing cellular senescence. The study is written clearly, contains several interesting insights that are important to the field and reveals a so far unacknowledged role of Glcd and the GCS in stem cell biology.

Response: We appreciate the reviewer for the comments that well summarized the major findings and significance of our study.

Major comments:

Fig2

A

The Western Blot shows an induction of GLDC expression upon Sox2 and Lin28A expression. Moreover, it seems that also Klf4 (probably mislabeled as Kif4 in the Figure) has the same effect. Could the authors please comment on that and if possible, provide a quantification of the band intensities across several experiments in order to be able to judge the effect size?

Response: Thanks for the concern and suggestion. We have corrected the typo in the revised manuscript. Following the reviewer's advice, we have quantified the band intensities value (IDV) of each band by the Image lab 3.0 software (Bio-Rad), which showed that all the six genes affected Glcd expression in varying degrees, but Sox2 and Lin28A upregulated Glcd much more significantly than other genes (**Figure R13A**, also as **Figure 2A** in the original manuscript). It is probable that these transcriptional factors can induce the expression of each other, and influence the expression of Glcd directly or indirectly.

Figure R13. Sox2 and Lin28A cooperatively controlled Gldc expression. (A) qRT-PCR and western blot analysis of the expression of Gldc in MEF cells overexpressing Oct4, Sox2, Klf4, c-Myc, Nanog and Lin28A individually. ACTIN served as the loading control. The data were presented as the mean \pm SD of three independent experiments. * $P < 0.05$ compared with the EV control. (Also as Figure 2A in the original manuscript)

M

The effect of Sox2 and Lin28A single knock downs on GLDC expression seems to be very mild, especially in comparison to experiments shown in 2B and 2F, again a quantification would help.

Response: Thank you for the helpful advice. We repeated the experiments and provided the results of three independent experiments here with quantification of the band intensities by the Image lab 3.0 software (Bio-Rad). The results showed that knockdown of either Sox2 or Lin28A significantly decreased the expression of Gldc. (Figure R14A, B). To avoid further confusion, we replaced this figure in the revised manuscript (Figure 2M).

Figure R14. Sox2 and Lin28A regulate Glc expression cooperatively. (A) V6.5 cells were infected with viruses expressing shSox2 or shLin28A or both of them, followed by analysis of Glc expression by western blotting. ACTIN served as the loading control (Also as Figure 2B, 2F, 2M in the original manuscript). **(B)** V6.5 cells were infected with viruses expressing shSox2 or shLin28A or both of them, followed by analysis of Glc expression by western blotting. ACTIN served as the loading control. (Also as Figure 2M in the revised manuscript)

In general, this figure provides evidence of transcriptional (Sox2) and posttranscriptional (Lin28A) control of GLDC expression. However, in 1G the authors show and also explicitly state in the result section of the text that transcriptional induction of GLDC during reprogramming precedes induction of 4F including Sox2. The authors should discuss the apparent discrepancy and if possible, formulate alternative hypothesis for Sox2-independent GLDC transcriptional control.

Response: Thank you for this important and relevant question. Firstly, transcriptional induction of Glc was observed during Yamanaka Factors-induced somatic reprogramming, even earlier than the induction of endogenous pluripotency genes (Figure 1). During the somatic reprogramming, exogenous Oct4, Sox2, Klf4 and cMyc were forced expressed in MEFs, and the expression of Glc was upregulated by exogenous pluripotent genes. To obtain a better understanding of the mechanism by

which Gldc is induced in somatic reprogramming, we forced expressed the factors used to induce iPS cell generation (Oct4, Sox2, Klf4, cMyc, Nanog or Lin28A), respectively, in MEFs, and found that overexpression of Sox2 and Lin28A could significantly increase the expression level of Gldc (**Figure 2**). The induction of Gldc was found to stimulate one-carbon metabolism that which is critical for deposition of H3K4me3 and the expression of pluripotent genes (**Figure 3**). Indeed, as shown in **Figure 2A**, we found other tested factors could mildly induce the expression of Gldc, especially Klf4, which is consistent with the previous report⁵. As we know, these pluripotent factors can induce the expression of each other in somatic reprogramming^{6, 7}, and influence the expression of Gldc directly or indirectly. We thank the reviewer for insightful comments that help us improve the current study. We have discussed this point in the Results section (**Figure 2**) of the revised manuscript.

Fig4

A and D

The representation of the GSEA needs improvement since the current figure is not readable at the presented resolution. Also, it would be helpful to display the total number of genes / gene sets analyzed and housekeeping pathways that may not change as controls.

Response: We appreciate the reviewer for the helpful advice and have improved the representation of the GSEA in the revised manuscript accordingly (**Figure R15A, R15B**, also as **Figure 4A, 4D** in revised manuscript). Moreover, we have included the GSEA of housekeeping pathway as controls in **Supplementary Figure 4A** in the revised manuscript (**Figure R15C**), and the data showed that the expression levels of genes in housekeeping pathway had no change in Gldc-knockdown cells compared to that in the NTC.

Figure R15. Knockdown of Gldc downregulated stem cell pluripotency and one carbon metabolism (A) GSEA of a gene set comprising stem cell pluripotency genes in shGldc- versus NTC-transfected V6.5 cells. (Also as Figure 4A in the revised manuscript) **(B)** GSEA of a gene set comprising genes related to one-carbon metabolism in shGldc- versus NTC-transfected V6.5 cells. (Also as Figure 4D in the revised manuscript) **(C)** GSEA of a gene set comprising genes related to housekeeping in shGldc- versus NTC-transfected V6.5 cells. (Also as Supplementary Figure 4A in the revised manuscript)

B

The panel shows that pluripotency marker gene expression is lost upon reduced Gldc expression. How specific is this effect? Are markers for different germ layers induced, are there any molecular or morphological hints on into what cellular state these cells differentiate into?

Response: Thank you for this important question. The Referee #1 has also raised the similar concerns. To investigate whether knockdown of Gldc in pluripotent stem cells

would induce differentiation, the mRNA levels of markers for differentiation were analyzed, and the results showed that the genes associated with endodermal differentiation, such as Gata3 and Wnt2, were upregulated in Gldc-knockdown V6.5 cells (**Figure R4A**). Indeed, it is an interesting question to make clear what cellular state these cells differentiate into and its underlying mechanism, which warrants future and independent studies.

Figure R4. The effect of Gldc knockdown on the expression of differentiating genes. (A) qRT–PCR analyses of the regulation of differentiating genes by Gldc in V6.5 cells. The data were presented as the mean \pm SD of three independent experiments. *P<0.05 compared with the NTC.

The downregulation of SOX2 upon Gldc knock down and opposite behavior upon overexpression suggests a positive feedback loop between these factors. Could the authors please comment on that possibility and how it fits in their proposed model Fig7?

Response: We truly appreciate our reviewer for the very insightful comments and agree completely with this opinion. In pluripotent stem cells and during somatic reprogramming, the upregulated Sox2 transcriptionally induced the expression of Gldc (**Figure R16A, R16B**, also as **Figure 2A, 2B** in the original manuscript) which promotes the one-carbon metabolism to maintain H3K4me3 on pluripotency related genes including Sox2 (**Figure R16C, R16D**, also as **Figure 4B, 4C** in the original manuscript). This suggests a positive feedback loop between Gldc and Sox2, which is critical for the acquisition and maintenance of pluripotency. Thank you for the interesting point and we have discussed this point in the Results section (**Figure 4**) in

the revised manuscript.

Figure R16. A positive feedback loop between Sox2 and Gldc in ES cells and during somatic reprogramming. (A) qRT-PCR and western blot analysis of the expression of Gldc in MEF cells overexpressing Oct4, Sox2, Klf4, c-Myc, Nanog and Lin28A individually. ACTIN served as the loading control. The data were presented as the mean \pm SD of three independent experiments. * P <0.05 compared with the EV control. (Figure 2A in the original manuscript) (B) qRT-PCR and western blot analysis of the expression of Gldc in V6.5 cells stably expressing shSox2 or the NTC. The data were presented as the mean \pm SD of three independent experiments. * P <0.05 compared with the NTC. ACTIN served as the loading control. (Figure 2B in the original manuscript) (C, D) qRT-PCR (C) and western blot (D) analyses of the regulation of pluripotency-related genes by Gldc in V6.5 cells. ACTIN served as the loading control. The data were presented as the mean \pm SD of three independent experiments. * P <0.05 compared with the NTC. (Figure 4B,4C in the original manuscript)

F

The pluripotency marker genes tested for H3K4me3 tested in Figure 4F suggest an expression and cell identity change due to a reduction of methylation levels of about 50 % in a Gldc knock down. Do the authors have control data from non-pluripotent associated genes or, ideally, could they provide genome-wide ChiP-Seq data to understand how selective the proposed action of the GCS is? Is it possible that this effect is a side-effect of the proposed anti-senescence activity observed in data of Fig5?

Given that conclusion and model of this work are largely based on this functional connection it deserves further investigation.

Response: We appreciate the reviewer for the constructive comments. To address this point, we repeated the ChIP experiments and added beta-Actin as non-pluripotent associated gene control. The results showed that knockdown of *Gldc* significantly decreased H3K4me3 on pluripotent genes, but had no effect on beta-Actin (**Figure R17A**). Similar results were observed in previous studies that alternative histone methylation selectively influence the modification of specific genes but not all the genes in embryonic stem cells^{8, 9}, however, the detailed mechanism is still under investigation¹⁰. To address whether the epigenetic changes is a side-effect of senescence, we added MG in the culture medium of V6.5 cells to induce senescence and tested the levels of H3K4me3. No significant changes in level of H3K4me3 was observed (**Figure R8A**, the Referee #1 raised similar concerns, for your convenience, we append the figures here once again), which suggested that *Gldc* regulates the pluripotent states via preventing senescence and maintaining histone modification independently.

Figure R17. *Gldc* knockdown reduced H3K4me3 levels on the transcription starting site of pluripotency genes. (A) ChIP-qPCR analysis of H3K4me3 enrichment in the promoters of *Pou5f1*, *Sox2*, *Klf4*, *Nanog* and *Actb* in V6.5 cells expressing sh*Gldc* or the NTC. The data were presented as the mean \pm SD of three independent experiments relative to the input. *Actb* served as the negative control. * $P < 0.05$ compared with the NTC.

Figure R8. Addition of MG had no effect on H3K4me3 level. (A) Western blot analysis of H3K4me3 modifications in V6.5 cells treated with various concentration gradients of MG. ACTIN served as the loading control.

I

What is the effect of formate only on these genes? This experiment might reveal if formate is rate limiting in the proposed network and might explain the observed overshoot for Klf4.

Response: We thank the reviewer for the helpful advice. We performed additional experiments to address this point. The mES cells V6.5 were cultured with the supplementation of formate and the mRNA levels of pluripotency-associated genes were analyzed. As a result, supplementation of formate increased the expression levels of pluripotent genes (**Figure R18A**), which may suggest that the observed overshoot for Klf4 was caused by formate itself.

Figure R18. Formate partially elevates the expression of pluripotency genes. (A) qRT-PCR analysis of the expression of Pou5f1, Sox2, Klf4 and Nanog in V6.5 with the supplementation of 1 mM formic acid for 48 hours. The data were presented as the mean \pm SD of three independent experiments.

Fig5D

How do the used MG concentrations compare to endogenously observed intracellular concentrations?

Response: Thank you for this point. The endogenously observed intracellular concentrations of MG were varying from 20 μ M to 40 μ M in different types of the cells¹¹. The experiments with MG treatment in our experiments were carefully performed referring to the previous studies^{12,13}.

Minor comments

Could the authors discuss how the observed changes in Gldc expression levels compare to differential expression during mouse embryogenesis?

Response: Thank you for the important and relevant question. To address this point, we analyzed public available RNA-seq data of Gldc expression during mouse embryogenesis (E-MTAB-6798). As a result, the expression of Gldc was gradually downregulated during the development of brain, heart, kidney, ovary and testis in mouse embryos (**Figure R19A**). It is consistent with our results that Gldc has high expression level in pluripotent stem cells, and is downregulated in somatic cells (**Figure 1E, 1F**). These data suggested that Gldc expression was correlated with pluripotent state.

Figure R19. The expression of Gldc during the development of five major organs of mouse. (A) RNA-seq analysis of the expression of Gldc during the development of five major organs of mouse embryos (E-MTAB-6798). E: embryonic day; P: postnatal day. The data are presented as the mean \pm SD of four independent experiments.

References

1. Wang, J. *et al.* Dependence of mouse embryonic stem cells on threonine catabolism. *Science (New York, N.Y.)* **325**, 435-439 (2009).
2. Labuschagne, C.F., van den Broek, N.J., Mackay, G.M., Vousden, K.H. & Maddocks, O.D. Serine, but not glycine, supports one-carbon metabolism and proliferation of cancer cells. *Cell Rep* **7**, 1248-1258 (2014).
3. Ron-Harel, N. *et al.* Mitochondrial Biogenesis and Proteome Remodeling Promote One-Carbon Metabolism for T Cell Activation. *Cell Metabolism* **24**, 104-117 (2016).
4. Sejersen, H. & Rattan, S.I. Dicarboxyl-induced accelerated aging in vitro in human skin fibroblasts. *Biogerontology* **10**, 203-211 (2009).
5. Kang, P.J. *et al.* Glycine decarboxylase regulates the maintenance and induction of pluripotency via metabolic control. *Metab Eng* **53**, 35-47 (2019).
6. Kim, J., Chu, J., Shen, X., Wang, J. & Orkin, S.H. An Extended Transcriptional Network for Pluripotency of Embryonic Stem Cells. *Cell* **132**, 1049-1061 (2008).
7. Neph, S. *et al.* Circuitry and dynamics of human transcription factor regulatory networks. *Cell* **150**, 1274-1286 (2012).
8. Lee, Y.H. *et al.* Protein arginine methyltransferase 6 regulates embryonic stem cell identity. *Stem Cells Dev* **21**, 2613-2622 (2012).
9. Wan, X. *et al.* Mll2 controls cardiac lineage differentiation of mouse embryonic stem cells by promoting H3K4me3 deposition at cardiac-specific genes. *Stem Cell Rev* **10**, 643-652 (2014).
10. Jambhekar, A., Dhall, A. & Shi, Y. Roles and regulation of histone methylation in animal development. *Nat Rev Mol Cell Biol* (2019).
11. Lee, D.Y. & Chang, G.D. Methylglyoxal in cells elicits a negative feedback loop entailing transglutaminase 2 and glyoxalase 1. *Redox Biol* **2**, 196-205 (2014).
12. Nokin, M.J. *et al.* Methylglyoxal, a glycolysis side-product, induces Hsp90 glycation and YAP-mediated tumor growth and metastasis. *Elife* **5** (2016).
13. Figarola, J.L., Singhal, J., Rahbar, S., Awasthi, S. & Singhal, S.S. LR-90 prevents methylglyoxal-induced oxidative stress and apoptosis in human endothelial cells. *Apoptosis* **19**, 776-788 (2014).

September 11, 2019

RE: Life Science Alliance Manuscript #LSA-2019-00413-TR

Prof. Ping Gao

University of Science and Technology of China

School of Life Sciences

University of Science and Technology of China, No.96, JinZhai Road Baohe District ,Hefei,Anhui,
230027,P.R.China

Hefei, Anhui 230026

China

Dear Dr. Gao,

Thank you for submitting your revised manuscript entitled "Glycine Cleavage System Determines the Pluripotency via Senescence and Epigenetic Regulation". As you will see, reviewer #2 re-assessed your work and appreciates the introduced changes, and we would thus be happy to publish your paper in Life Science Alliance pending final revisions necessary to meet our formatting guidelines.

- please deposit your RNA-seq data and provide the accession code in the material and methods section
- wherever p values are listed, please also indicate in the figure legends which statistical test was used
- please link your ORCID iD to your profile in our submission system
- please upload the supplementary figures as individual files and the supplementary tables as either word docx or excel files
- please add a callout to Fig 7 into the manuscript text
- please make the scale bars in Fig3J more visible

A. FINAL FILES:

- An editable version of the final text (.DOC or .DOCX) is needed for copyediting (no PDFs).

B. MANUSCRIPT ORGANIZATION AND FORMATTING:

Sincerely,

Andrea Leibfried, PhD
Executive Editor
Life Science Alliance
Meyerhofstr. 1

69117 Heidelberg, Germany
t +49 6221 8891 502
e a.leibfried@life-science-alliance.org
www.life-science-alliance.org

Reviewer #2 (Comments to the Authors (Required)):

The authors significantly improved the manuscript and addressed the raised questions appropriately. In my opinion, the manuscript can now be accepted for publication.

September 17, 2019

RE: Life Science Alliance Manuscript #LSA-2019-00413-TRR

Prof. Ping Gao

University of Science and Technology of China

School of Life Sciences

University of Science and Technology of China, No.96, JinZhai Road Baohe District ,Hefei,Anhui,
230027,P.R.China

Hefei, Anhui 230026

China

Dear Dr. Gao,

Thank you for submitting your Research Article entitled "Glycine Cleavage System Determines the Pluripotency via Senescence and Epigenetic Regulation". It is a pleasure to let you know that your manuscript is now accepted for publication in Life Science Alliance. Congratulations on this interesting work.

DISTRIBUTION OF MATERIALS:

Again, congratulations on a very nice paper. I hope you found the review process to be constructive and are pleased with how the manuscript was handled editorially. We look forward to future exciting

submissions from your lab.

Sincerely,
